# Daily Profile of miRNAs in the Rat Colon and In Silico Analysis of Their Possible Relationship to Colorectal Cancer

**DOI:** 10.3390/biomedicines13081865

**Published:** 2025-07-31

**Authors:** Iveta Herichová, Denisa Vanátová, Richard Reis, Katarína Stebelová, Lucia Olexová, Martina Morová, Adhideb Ghosh, Miroslav Baláž, Peter Štefánik, Lucia Kršková

**Affiliations:** 1Department of Animal Physiology and Ethology, Faculty of Natural Sciences, Comenius University in Bratislava, 842 15 Bratislava, Slovakia; vanatova2@uniba.sk (D.V.); katarina.stebelova@uniba.sk (K.S.); lucia.olexova@uniba.sk (L.O.); morova4@uniba.sk (M.M.); miroslav.balaz@savba.sk (M.B.); peter.stefanik@uniba.sk (P.Š.); lucia.krskova@uniba.sk (L.K.); 2First Surgery Department, University Hospital, Comenius University in Bratislava, 811 07 Bratislava, Slovakia; reis1@uniba.sk; 3Institute of Food, Nutrition and Health, ETH Zürich, 8603 Schwerzenbach, Switzerland; adhideb.ghosh@hest.ethz.ch; 4Biomedical Research Center, Slovak Academy of Sciences, 845 05 Bratislava, Slovakia

**Keywords:** miR-150, miR-30d, miR-142, *bcl2*, *myb*, *cry1*, circadian, DGCR8, age-dependent

## Abstract

**Background:** Colorectal cancer (CRC) is strongly influenced by miRNAs as well as the circadian system. **Methods:** High-throughput sequencing of miRNAs expressed in the rat colon during 24 h light (L)/dark (D) cycle was performed to identify rhythmically expressed miRNAs. The role of miR-150-5p in CRC progression was analyzed in DLD1 cell line and human CRC tissues. **Results:** Nearly 10% of mature miRNAs showed a daily rhythm in expression. A peak of miRNAs’ levels was in most cases observed during the first half of the D phase of the LD cycle. The highest amplitude was detected in expression of miR-150-5p and miR-142-3p. In the L phase of the LD cycle, the maximum in miR-30d-5p expression was detected. Gene ontology enrichment analysis revealed that genes interfering with miRNAs with peak expression during the D phase influence apoptosis, angiogenesis, the immune system, and EGF and TGF-beta signaling. Rhythm in miR-150-5p, miR-142-3p, and miR-30d-5p expression was confirmed by real-time PCR. Oncogenes *bcl2* and *myb* and clock gene *cry1* were identified as miR-150-5p targets. miR-150-5p administration promoted camptothecin-induced apoptosis. Expression of *myb* showed a rhythmic profile in DLD1 cells with inverted acrophase with respect to miR-150-5p. miR-150-5p was decreased in cancer compared to adjacent tissue in CRC patients. Decrease in miR-150-5p was age dependent. Older patients with lower expression of miR-150-5p and higher expression of *cry1* showed worse survival in comparison with younger patients. **Conclusions:** miRNA signaling differs between the L and D phases of the LD cycle. miR-150-5p, targeting *myb*, *bcl2*, and *cry1*, can influence CRC progression in a phase-dependent manner.

## 1. Introduction

Rhythmic gene expression with a period approximating a 24 h (h) day (L)/night (D) cycle is under the regulatory influence of the circadian system. The circadian system is hierarchically organized. It is composed of a central oscillator localized in the suprachiasmatic nuclei of the hypothalamus (SCN) and peripheral oscillators present in most human tissues including the colon [1]. Although oscillations in gene expression are autonomous, they are, under in vivo conditions, harmonized by the SCN to cope with the LD cycle [2].

Circadian oscillations are triggered by a molecular feedback loop composed of clock genes and transcription factors. The most important transcription factors involved in the generation of circadian rhythms are BMAL1 and CLOCK, which induce the expression of clock genes *per* and *cry* via enhancer E-box. Protein products of *per* and *cry* genes possess the capacity to regulate their own transcription by interfering with E-box-mediated induction executed by BMAL1/CLOCK. The basic circadian loop is further influenced by additional transcriptional factors and modifying enzymes that generate a delay between the induction and inhibition of clock gene expression [1]. This chain of molecular events leads to the generation of oscillations with approximately 24 h lasting period [2].

E-box is an abundantly expressed regulatory region, and except for clock gene, it is also used to convey circadian regulation to many other genes [3]. As a result, approximately 8–10% of the transcriptome is expressed rhythmically. However, the pattern of genes exerting rhythm in expression shows remarkable tissue specificity. Differences are observed not only in the number of rhythmic genes relative to the overall transcriptome but also in the amplitude and acrophases of particular rhythms [4,5].

Although the regulatory relationship of clock genes with respect to mRNA regulation is well documented [5], far less is known about circadian regulation of small non-coding RNA (miRNA) [6,7]. Biosynthesis of miRNA begins, similarly to mRNA, with transcription of the primary transcript of miRNA (pri-miRNA) from DNA. Further steps of biosynthesis, however, differ between miRNA and mRNA. pri-miRNAs are processed in the nucleus by an enzymatic complex containing Drosha and DGCR8 that cleaves the miRNA hairpin and generates precursor miRNAs (pre-miRNAs). After pre-miRNAs export from the nucleus, they are processed by the endonuclease Dicer, which, by cutting the terminal loop, generates mature miRNAs. Mature miRNAs consist of guide and passenger strands that usually strongly differ in their concentrations within the cell [8]. miRNA biosynthesis can be regulated at the level of pri-miRNA, pre-miRNA, and mature miRNA. As E-box-mediated regulation influences the very first step of biosynthesis, it is expected that the circadian rhythmic pattern of pri-miRNA expression can be erased by consecutive steps of biosynthesis [9]. Moreover, as sequences of pri-miRNAs are not annotated in the database yet, it is not easy to inspect the presence of E-box in their sequence.

It has been reported that non-coding RNAs (ncRNAs) show a similar proportion of rhythmically expressed genes like protein-coding genes [5]. In particular, a daily rhythm in the expression of miR-96 and mir-182 has been observed in the mice retina [10]; rhythm in the expression of miR-17, miR-212, miR-219, and miR-132 has been detected in mice SCN [11,12,13]; miR-132 exerts a daily rhythm in the hippocampus of mice [14]; miR-20a, miR-141, and miR-16 show a rhythmic pattern during the 24 h cycle in the small intestine of rat [15]; and miR-142 exerts a rhythmic pattern in the cell culture of NIH3T3 and 293ET cells following serum shock [16].

In the mouse liver, 13.3% of miRNAs showed a rhythmic pattern under synchronized LD conditions. miRNAs differed in the acrophases of their expression, and the strongest cluster of rhythmic miRNA was composed of those with maximal values in the second half of the L phase of the LD cycle, which included miR-142-5p, miR-150-5p, miR-92, miR-17, miR-20b, and others [17]. There was a daily pattern showing significant rhythm exerted by 53 miRNAs, including let-7g-3p, miR-142-5p, miR-150-5p, and miR-185-5p, in the mouse liver, according to Vollmers et al. [18]. Similarly, the presence of 57 oscillating primary transcripts (out of 558 miRNA transcripts) has been reported in the same tissue by Wang et al. [9]. Liver highly enriched miR-122 shows a distinct daily rhythm in this tissue [19]. The rhythmic pattern of miRNAs with BMAL1 binding site (pri-mir-23b~27b~24-1, pri-mir-101a, and pri-mir-378) was weakened or diminished in liver-specific *bmal1* knockout mice. Although the rhythmic pattern of mature miRNAs showed a tendency to dampen compared to pri-miRNAs, miR-24-3p, miR-101a-3p, miR-378-3p, and miR-122-3p exerted a rhythmic expression. From 194 miRNA transcripts, 57 exerted a different pattern or expression in ClockΔ19 mice [20]. A daily rhythm in pre-miR-34a has been detected in the liver and heart [21] and prefrontal cortex of rats [22]. pre-miR-30c exerted a rhythmic pattern in the liver, kidney, and heart of rats [21].

Human cell lines MCF-10A, MCF-7, and/or MDA-MB-231 exerted a daily rhythm in the expression of miR-141-5p, miR-1225-5p, miR-17-5p, miR-222-5p, miR-769-3p, and miR-548ay-3p [23]. A rhythmic pattern in miR-27b, miR-16, and miR-181 [24,25] has been detected in human leukocytes. In the plasma of healthy male volunteers, 26 out of 79 measurable miRNAs exerted a rhythm in their levels. Among miRNAs showing a rhythmic pattern were miR-150-5p, 363-5p, 24-3p, 15b-5p, 139-5p, and 34a-5p [26].

Under physiological conditions, miRNAs strongly contribute to gastrointestinal homeostasis maintenance [27]. In *dicer*-deficient mice, pronounced changes in intestinal epithelial morphology have been reported, including epithelial disorganization, decreased number of goblet cells, and an increase in apoptosis in crypts and cell migration. Dicer deletion also caused impaired mucus generation and an increase in gut permeability. A regulatory role of miR-192 has been implicated in this respect [28]. Overexpression of some miRNAs has also been reported to influence gut function, e.g., miR-122 increases intestinal permeability by targeting the gene coding occludin [29]. Similarly, it was shown that silencing of miR-21 improves susceptibility to experimentally induced colitis [30].

The positive or negative role of miRNAs in gut physiology strongly depends on their interaction with the epithelial barrier (EB). miRNAs can interact with the EB directly, via interaction with the structural molecules of the EB [29], and indirectly, via their interaction with the microbiome [30]. miR-21, miR-122, miR-142, miR-34a, miR-155, and others are known to influence the expression of occludin and claudins [31]. miRNAs influencing the intestinal microbiome include miR-155, miR-150, miR-143, miR-148a, miR-200c, miR-26b, miR-146a, miR-18a, and others [32].

Another way that miRNAs influence gut physiology is interaction with the immune system [27], e.g., miR-155-deficient mice are more prone to pathogen infection in the gut because of the deregulated function of gut B cells [33].

An effect of miRNA on intestinal physiology has also been reported with respect to the regulation of proliferation, differentiation, apoptosis, and autophagy [27]. The intestine of miR-143/145 knockout mice showed decreased regeneration capacity after injury [34]. It has also been shown that the miR-30 family regulates proliferation and differentiation of intestinal epithelial cells [35]. miR-142-3p inhibits autophagy by inhibition of ATG16L1 and induces apoptosis. These changes led to a decrease in cell survival under starvation condition [36].

The contribution of miRNA to cell cycle control is enormous. miRNAs in most cases inhibit gene expression. Their effect can be either oncogenic or oncostatic depending on target genes [37]. Research focused on translational medicine has demonstrated the regulatory role of miRNAs in the progression of many types of solid cancers [38]. Molecules mimicking miRNA action or miRNA inhibitors or sponges are being tested with respect to all aspects of cancer progression and treatment.

The best-known oncogenic miRNAs include miR-21, miR-155, and miRNAs from the miR-17-92-cluster. miR-21 promotes cancer progression by inhibition of tumor suppressors PTEN [39] and Pdcd4 [40]. miR-155 has been shown promote cancer progression by inhibition of WEE1 and PTEN [41], and its effect has been tested in a clinical trial focused on treatment of cutaneous T-cell lymphoma [42]. Several members of the miR-17-92 cluster play important roles in tumorigenesis by targeting genes coding p21 and PTEN [43], Bim, EGR2, AIBI [44], RDN3 [45], and others [46].

The best known miRNAs with prevailing tumor suppressor activity include members of the let-7 family that inhibit expression of oncogenes RAS [47], MYC [48], HMGA2 [49], and STAT3 [50]. miR-34a, whose expression is induced by p53, inhibits expression of cancer-promoting MYC [51] and Bax [52] and in this way contributes to p53’s oncostatic effect. miR-150 inhibits expression of oncogenes Myb [53] and HMGA2 [54] and apoptosis inhibitor Bcl2 [55]. miR-150 also inactivates the Wnt/β-catenin pathway by inhibition of β-catenin expression [56].

Although there are some miRNAs that are taken as unambiguous with respect to cancer progression, in most cases a cumulative effect of miRNAs on the transcriptome and/or specific tissue must be considered when they are associated with oncogenesis, e.g., miR-142-3p [57], miR-30d [58], miR-185 [59], miR-425 [60], miR-129 [61], miR-29a [62], etc.

CRC is among the most frequently diagnosed reasons for cancer mortality worldwide [63]. Understanding of CRC development is complicated by its long-time asymptomatic period and the influence of numerous risk factors and comorbidities [64,65,66]. Factors that have been associated with outcomes of CRC patients include the expression of clock genes [7,67,68,69,70,71,72]. This is not surprising considering the robust influence of the circadian system on cell cycle progression [73,74]. Several clock genes have also been linked to altered CRC progression either as tumor suppressors (e.g., *per2*) or oncogenes (e.g., *cry1*) [75,76]. The circadian system shapes the expression of key cell cycle regulators such as cyclines B1, D1, E, and A and cell cycle regulators *wee1* and *c-myc* [77,78,79,80].

There is huge supporting evidence connecting cancer progression with the circadian system and miRNAs. Evidence supporting the impact of miRNA peak time on oncogenesis is also available. A role of miR-21 in the circadian regulation of apoptosis was demonstrated in a model of *Apoe*^−/−^*Mir21*^−/−^ mice [81]. It has also been demonstrated that expression of miR-21 is light inducible, and that the light-elicited effect of miR-21 was abolished in miR-21-/-mice [82]. Moreover, the expression of miR-21 exerts a daily rhythm in atherosclerotic lesions [81] and in the heart [82]. Circadian disruption has led to deregulation of miRNA expression in mammary tissue and downregulation of miR-127 and miR-146b, which were linked to increased activity of STAT3 and BCL6 [83]. Regulatory interactions of miRNA and the circadian system have been addressed by in silico analysis showing that defined set miRNAs with sex-dependent expression can interact with clock gene *per2* and *cry2*-mediated regulation of PTEN and p53 [84]. Both PTEN and p53 are well known for their role in CRC onset [39,85]. miR-122 with rhythmic expression in the liver modulates chromatin remodeling [19].

As colorectal cancer progression is strongly influenced by miRNAs [86] as well as the circadian system [75], we performed screening of miRNAs expressed in the rat colon during a 24 h cycle under synchronized conditions to identify rhythmically expressed miRNAs. Abundant miRNAs with rhythmic patterns were sorted according to their acrophase into four clusters covering the whole 24 h cycle. A search for target genes of rhythmic miRNA grouped according to the peak of their expression was performed with the use of the miRTarBase database, which collects information about experimentally validated miRNA:mRNA interactions [87]. Gene ontology enrichment analysis (GO) of target genes was performed with the use of the Panther database [88]. The rhythmic profile of selected miRNAs was validated by real-time PCR in the rat colon (miR-150-5p, miR-142-3p, and miR-30d-5p) and DLD1 cells (miR-150-5p). The effect of miR-150-5p administration on apoptosis, wound healing, and rate of metabolism was tested in DLD1 cells and compared with the results of GO analysis. Attention was also paid to the reciprocal influence of miR-150-5p on clock gene expression. The results of next-generation sequencing (NGS) screening, GO analysis, and in vitro studies were compared with the expression of candidate genes and survival in human CRC patients.

## 2. Materials and Methods

### 2.1. Patients

The cohort of patients consisted of 47 patients that underwent surgery to remove tumor tissue and were not exposed to any previous treatment related to CRC. The average age of patients was 68.66 ± 1.82 years. Samples were taken during surgery from the tumor and proximal tissue (ascending part of the gut without signs of tissue transformation; ≥10 cm from the tumor). Tissue samples were collected in liquid nitrogen and then stored at −70 °C until RNA extraction. Appendix A shows detailed information about the cohort.

### 2.2. In Vitro Experiments

To evaluate the effect of miR-150-5p on apoptotic intensity, the human colorectal carcinoma cell line DLD1 (ATCC, Manassas, VA, USA) was used. Cells were cultured in RPMI 1640 GlutaMax medium (Gibco, Grand Island, NY, USA) with 10% FBS, 100 U/mL penicillin, and 100 μg/mL streptomycin (Biosera, Nuaille, France). Experiments were performed in a HF90 humidified incubator (Heal Force, Shanghai, China) with 5% CO_2_ and 37 °C. Cells cultured in marginal rows and columns of the 96-well plate were not used in experiments.

To measure cell response to miR-150-5p, miRIDIAN microRNA Human hsa-miR-150-5p mimic or mimic together with inhibitor at a concentration of 50 nM and DharmaFECT 1 Transfection Reagent (Horizon Discovery, Cambridge, UK) were used. To exclude the effect of transfection on the cell culture, control wells were transfected with miRIDIAN microRNA Hairpin Inhibitor Negative Control #1 and/or miRIDIAN microRNA Mimic Negative Control #1 (Horizon Discovery, Cambridge, UK) in concentrations corresponding to oligos administered to experimental wells. To perform the experiment, cells were seeded in a 25 mL cell culture flask under the conditions described above. When cells were confluent, the culture was trypsinized, and cells were seeded on a plate with the density specified in the description of each particular experiment. Transfection was performed together with cell passaging. Oligos were added into cell culture immediately before seeding and were not washed away during the experiment. In all cases, the highest volume of transfection reagent from the range recommended by the manufacturer was used.

To test the effect of miR-150-5p on apoptosis, transfected cells were seeded on 96-well plate at a density of 4 × 10^4^ cells/well. The effect of miR-150-5p was tested alone or in combination with the apoptosis inductor camptothecin (Bio-Connect, Huissen, The Netherlands), used in a 5.7 μM concentration. Camptothecin was dissolved in DMSO to obtain 2 mg/mL stock solution, which was diluted to achieve the desired working concentration. When camptothecin was not administered, a corresponding volume of DMSO was administered to the cell culture. When the effect of miR-150-5p was tested in combination with camptothecin, camptothecin (or DMSO) was administered 4 h later. Apoptotic cells were detected after 24 and 48 h by Invitrogen™ CellEvent™ Caspase-3/7 Detection Reagent (Thermo Fisher Scientific, Waltham, MA, USA) with 5 μM concentration. Fluorescent signal was detected with the use of inverse fluorescent microscope NIB-100F (NOVEL, Ningbo, China) and BEL Capture 3.2 software (Ningbo, China). The intensity of fluorescence was measured by ImageJ 1.53a (NIH, Bethesda, MD, USA).

To measure clock gene expression in response to miR-150-5p administration, DLD1 cells were seeded in a 24-well plate at a density 1 × 10^5^ cells/well. miR-150-5p mimic, inhibitor, or control oligos dissolved in medium and DharmaFECT 1 Transfection Reagent (Horizon Discovery, Cambridge, UK) were added to the cell culture during the last step of cell passaging as described above. Plates were incubated at 37 °C and 5% CO_2_ for 72 h. After this time, samples were collected into 250 μL RNAzol RT (Molecular Research Center, Cincinnati, OH, USA) and stored under −70 °C until RNA extraction. Each well was analyzed separately, and each group consisted of six wells.

To measure the daily pattern in gene expression, DLD1 cells were seeded on six well plates at a density of 3 × 10^5^ cells/well using the medium described above. On day five of incubation, the culture medium was replaced by FBS serum-free medium for 6 h. Afterwards, cells were synchronized by serum shock. Sample collection was initiated 6 h after the serum shock in 6 h intervals (*n* = 4–5) into 400 μL of RNAzol RT. Samples were stored under −70 °C until RNA extraction.

Scratch assay was used to test the effect of miR-150-5p on wound healing in DLD1 cells, which were cultured and transfected by miR-150-5p mimic as described above in 96-well plates. When the cell culture reached a confluence, the monolayers were scratched using a 10 μL sterile tip. Pictures were taken immediately after wound generation and 24 and 48 h later. Wound closure was documented by an inverted fluorescence microscope NIB-100F and analyzed by ImageJ 1.53a (NIH, Bethesda, MD, USA). Each group consisted of six wells.

The effect of miR-150-5p mimic and/or inhibitor was evaluated by (CellTiter 96 AQueous Cell Proliferation Assay, Promega, Madison, WI, USA) employing the modified tetrazolium compound (MTS), whose conversion reflects the activity of mitochondrial dehydrogenase producing NADPH or NADH in metabolically active cells. Cells were cultured and transfected in 96-well plates as described above. Each group consisted of 10 wells. Tetrazolium compound was mixed with phenazine ethosulfate in a ratio of 20:1 immediately before use, and 20 μL of mixture was added to each well containing cells intended for experiment under conditions of dim light. After reaction, initiation cells were cultured in the dark, at 37 °C and 5% CO_2_. After 3 h of incubation, absorbance was measured at 490 nm using a UV spectrophotometer (Epoch, Agilent Technologies, Inc., Santa Clara, CA, USA).

### 2.3. Animal Study

Adult male Wistar rats (*n* = 25, weight 435.12 ± 0.5 g) were acquired from a breeding facility (Dobra Voda, SAV, Slovakia). During the acclimatization period and experiment they were provided with food and water ad libitum and kept under conditions of L:D cycle 12:12 h. Time is expressed in relative units—Zeitgeber time (ZT). The beginning of the L phase is defined as Zeitgeber time 0 (ZT0). After the acclimatization period, the experiment continued in the same laboratory under the same conditions. The room was dedicated only to this experiment. Only directly involved staff had access to the laboratory and experimental animals. Animals were weighed at the beginning and end of the experiment. Entrainment of all rats by the LD cycle was validated by activity monitoring as described previously [89].

Sampling of the colon was performed at 4 h intervals during the 24 h cycle (animals were selected randomly for each time point, ZT10, ZT14, ZT18, ZT22, ZT26 *n* = 4, ZT30 *n* = 5). Rats were anaesthetized with isoflurane and subsequently decapitated. A low-intensity red light was used for sample collection during the D phase. Samples of tissue were resected within 5 min, frozen in liquid nitrogen, and stored under −70 °C until RNA extraction.

The experimental protocol was approved and registered by the Ethical Committee for the Care and Use of Laboratory Animals at Comenius University in Bratislava and the State Veterinary Authority of Slovak Republic.

### 2.4. PCR

Large RNA and miRNA were extracted from tissues (*n* = 25, 4–5 animals for each time point) and cells using RNAzol, according to the manufacturer’s instructions (MRC, Inc., Houston, TX, USA; protocol for isolation of large RNA and small RNA fractions). To measure mRNA expression, 1 μg of large RNA fraction isolated from tissues and 0.3 μg from cells was used to synthesize cDNA with the ImProm-II Reverse Transcription System (Promega, Madison, WI, USA) and random hexamers, according to the manufacturer’s instructions.

The gene expression quantification was carried out by real-time PCR using the QuantiTect SYBR Green PCR kit (Qiagen, Hilden, Germany) and the StepOne™ Plus Real-Time PCR System thermocycler (Applied Biosystems, Waltham, MA, USA) with specific primers provided in the Appendix A. Real-time PCR conditions were as follows: hot start at 95 °C for 15 min followed by 40 cycles of 94 °C for 15 s (s), 53 °C for 30 s, and 72 °C for 30 s and melting curve analysis.

To synthesize cDNA for miRNA analysis, the miRCURY LNA RT Kit (Qiagen, Hilden, Germany) was used. Expression of miRNA was measured by miRCURY LNA SYBR Green PCR Kit (Qiagen, Hilden, Germany) using the following program: hot start at 95 °C for 2 min followed by 35–40 cycles of 95 °C for 10 s and 53–60 °C for 60 s. The last step of the program was melting curve analysis. Expression of miR-150-5p was measured by hsa-miR-150-5p miRCURY LNA miRNA PCR Assay (Qiagen, Hilden, Germany). Sequences of primers for miR-142-3p and miR-30d-5p are provided in Appendix A.

The expression of mRNA and miRNA was normalized with respect to the housekeeper(s) (*rnu6-1*, *rnu6-2*, *snord47*, and/or *β-actin*) that exerted the lowest responsiveness for that particular treatment. In real-time PCR, we perform arbitrary quantification with a standard curve specific to each particular tissue and gene (including housekeepers). Before the use of a gene for normalization, we check the sensitivity of housekeeper expression to treatment (and/or rhythmicity). This control is facilitated by StepOnePlus™ software v2.3 and chemistry, which allow the use of ROX fluorescent dye for normalization of a fluorescent signal for different plate wells to reduce the eventual heterogeneity of measurement. After quantification of housekeeper expression with the use of a calibration curve and appropriate statistical analysis, we can detect if housekeeper expression (specifically for each tissue and gene) changes with respect to treatment and/or LD cycle. Only the housekeeper that does not respond to treatment is used for normalization. Sequences of primers for housekeeper measurement can be found in Appendix A.

### 2.5. NGS

After passing the total RNA quality control, the NEBNext^®^ Small RNA Library Prep Set for Illumina was used for library preparation according to the manufacturer’s instructions (New England Biolabs, Ipswich, MA, USA). RNA sequencing was performed as single-end, with a 50 base read setting on the NovaSeq 6000 System (Illumina, San Diego, CA, USA). The miRNA sequencing data were processed using the ncPRO-seq pipeline v1.6.5 [90], which includes quality control, alignment with Bowtie [91] to the reference rat genome (UCSC rn5), expression quantification, and annotation. miRNA annotations were performed using miRBase v21, while other small non-coding RNAs were annotated using Rfam v11. All RNAseq data are available in Appendix A.

### 2.6. In Silico Analysis

The workflow of in silico analysis is shown in Appendix A. The major aim was to determine the regulatory pathways involved in oncogenesis that are influenced by miRNAs depending on time when miRNA expression reaches its maximum.

The expression of miRNAs and pre-miRNAs identified by screening was tested by cosinor analysis to select those with the rhythmic pattern. Based on cosinor analysis, miRNAs and pre-miRNAs were sorted into four clusters according to their acrophase: D1—maximum expression during the first half of the D phase of the LD cycle, D2—maximum expression during the second half of the D phase of the LD cycle; L1—maximum expression during the first half of the L phase of the LD cycle, and L2—maximum expression during the second half of the light phase of the LD cycle.

The first inclusion criterion was high expression of miRNA. miRNAs with rhythmic pattern were sorted according to the intensity of their expression based on quartiles determination. Only miRNAs with expression above the upper quartile (miRNAs with highest expression) were considered in further steps of analysis.

The second inclusion parameter was rhythmic expression of corresponding pre-miRNA with expression above the median.

Target genes of miRNAs fulfilling the inclusion criteria were determined with the use of the miRTarBase database, which collects information about experimentally validated miRNA:mRNA interactions [87]. Only interactions supported by strong evidence were included in the further analysis. Target genes were ascertained separately for clusters D1, D2, and L1 (the L2 cluster was empty as no miRNA peaked in the second half of the L phase).

With the use of Venn diagram analysis (https://bioinformatics.psb.ugent.be/webtools/Venn/ (accessed on 15 January 2025)), genes specific for miRNAs from the D (clusters D1 and D2) and L (cluster L1) phases were selected. Target genes specific to particular miRNA clusters were subjected to ontology analysis [88] to reveal which regulatory pathways are involved. Pathways linked to specific genes and miRNAs were analyzed by Venn diagram analysis to identify those that are regulated in acrophase-dependent manner.

### 2.7. Statistical Analysis

Sample laboratory processing was randomized. During extraction, samples selected for one set were mixed, to include samples from different time-points, different patients, different tissues (cancer vs. control), etc. Reverse transcription (RT) was organized in the same way, but the set of samples for RT always differed from that used during isolation. In real-time PCR, samples to be compared were measured with the same master mix. To eliminate the bath effect in the time-series experiment, the expression of clock genes *per2* and *bmal1* was measured (Appendix A). As clock genes *per2* and *bmal1* exert inverted acrophases in their mRNA daily pattern, their expression was used as an internal control.

Differences between two groups were compared by unpaired *t*-test. Correlation between mature and premature forms of miRNAs was determined by regression analysis.

To analyze the daily profile in locomotor activity and gene expression, cosinor analysis was used [92]. Data were fitted into a cosinor curve with 24 h period, and when experimental data significantly matched the cosinor curve, its parameters were calculated with 95% confidence limits: mesor (the time series mean), amplitude (one half of the peak-trough difference expressed herein relative to the mesor), and acrophase (peak time referenced to the time of lights on in the animal facility).

To evaluate the 5-year survival of patients in relation to the median miR-150-5p and *cry1* expression, a Kaplan–Meier survival curve and a log-rank test were used. The day of the surgery was the starting point for the log-rank test.

In the histograms and line graphs, data were provided as a mean ± standard error of the mean (SEM). Differences were considered statistically significant when the *p*-value < 0.05.

## 3. Results

To validate proper synchronization of animals to the external LD cycle, locomotor activity has been monitored during the experiment. There was a pronounced daily rhythm with high activity during the D phase and low activity during the L phase, as expected (cosinor, *p* < 0.05, Appendix A).

We observed a clear-cut daily rhythm in the 24 h profile of *per2*, *bmal1*, and *rev-erbα* expression. Expression of *per2* peaked at the beginning of the D phase, *bmal1* expression showed the highest level at the transition from D to L phase, and the acrophase of *rev-erbα* was at the end of the L phase (cosinor, *p* < 0.05, Appendix A).

Expression of enzymes involved in miRNA biosynthesis did not show significant rhythmicity in daily pattern of expression (cosinor); however, there was a trend to higher expression of *dgcr8* during the D phase compared to L phase (*t*-test, *p* = 0.065, Appendix A).

NGS sequencing revealed 371 mature miRNAs and 438 pre-miRNAs in the rat colon sampled during 24 h LD cycle (Figure 1A and Figure 1B, respectively). From the whole set of mature miRNAs, 199 miRNAs (54%) originated from one pre-miRNA and expressed both 3p and 5p strands. In addition, 126 mature miRNAs (34%) coming from one pre-miRNA were present only in one form (3p or 5p). In 35 cases (9%), mature miRNAs were present in 3p as well as 5p form and originated from several pre-miRNAs. In 11 miRNAs (3%), only one strand displayed measurable levels and came from several pre-miRNAs (Figure 1A).

Sequencing revealed that 369 pre-miRNAs (74%) come from one gene and express a mature form of miRNA. In 47 cases (23%), pre-miRNAs were coded by more than one gene, and a corresponding mature form was detected. Finally, expression of 14 pre-miRNAs (3%) was not accompanied by a mature form of miRNA. Expression of pre-miRNAs without a mature form present in sample was usually very low, with the exception of pre-miR-3575, which expressed a high level of its precursor form (Figure 1B).

A correlation between the guide and/or passenger strand of mature and corresponding pre-miRNAs coded by one gene was observed in 177 cases (89%). However, a significant correlation between both strands and pre-miRNA was observed only in 26 cases (13%). Expression of 5p but not 3p strand correlated with pre-miRNA in 77 cases (39%). Expression of 3p but not 5p strand was associated with expression of corresponding pre-miRNA in 74 cases (37%). The strand that correlated with pre-miRNA was usually a guide strand. When 5p but not 3p strand correlated with pre-miRNA, in 96% of cases the 5p strand was a guide strand. Similarly, when 3p but not 5p correlated with pre-miRNA, 3p was a guide strand in 99% of cases. Therefore, the proportion of 5p and 3p strands correlating with pre-mature miRNA was nearly equal. Expression of mature miRNAs with neither 5p nor 3p strand correlated with pre-miRNA was generally very low (Figure 2).

Mature and corresponding pre-miRNAs were more likely to correlate when expression of the 5p mature strand was high. When expression of the 3p strand was inspected, the correlation between the mature strand and corresponding pre-miRNA was less dependent on expression, although there was a prominent trend to increased expression in 3p strands exerting a correlation with corresponding pre-miRNA (Figure 2). This analysis included miRNAs that express both strands and are coded by one gene (54%).

When expression of 5p and 3p was sorted according to quartiles, there was a trend to higher expression of 5p guide strings compared to 3p guide strings (Figure 3).

Nearly 10% of identified miRNAs exerted a daily rhythm (cosinor, *p* < 0.05, Figure 4A). A list of miRNAs with significant rhythmic pattern sorted according to intensity of expression is provided in Table 1. The highest proportion of rhythmically expressed miRNAs was identified in the 75th percentile. The number of miRNAs in lower percentiles was similar (Table 1).

The first row shows the percentage of miRNAs in a particular cluster with respect to all detected mature miRNAs. In the case of pre-miRNAs, the number of sequences with rhythmic pattern (Figure 4B) showed even more pronounced differences among percentiles, with the highest proportion of rhythmic sequences in the 75th percentile (Table 2). Interestingly, not all rhythmically expressed mature miRNAs were positively associated with pre-miRNA showing oscillating levels.

Comparison of the miRNAs in Table 1 and Table 2 reveals that pre-miRNAs and mature miRNAs strongly parallel each other. In most cases, pre-miRNAs exerting a rhythm in expression during 24 h cycle were associated with rhythmically expressed mature miRNAs. pre-miR-219a-1 and miR-877 exerted a rhythmic profile at the level of pre-miRNA expression, and P did not reach the level of significance in mature miRNA expression. However, in the case of miR-219a-1, the daily pattern was very close to the significance cut off. Similarly, precursor forms of mature miRNAs miR-455-5p and miR-24-2-5p did not exert a significant rhythmic pattern; however, P was very close to the level of significance cut-off.

Comparison of miRNAs based on the ratio of expression during the D phase relative to L phase (D/L) of the 24 h cycle revealed that there is a pronounced trend to higher expression of miRNAs during the D phase compared to L phase (Figure 5A). Correlation of D/L ratio with intensity of expression indicates that the highest expressions show a tendency to tonic expression (*p* < 0.001, Figure 5B). There are more miRNAs showing increased expression during the D time compared to those with maximum expression exerted during the L phase of the LD cycle (Figure 5).

Based on a comparison of the results shown in Table 1 and Table 2 and results of cosinor analysis, miRNAs showing a rhythmic profile and high expression were selected and sorted according to the acrophase of the miRNA into four percentiles: D1 (ZT12–ZT18)—miR-128, miR-129, miR-139; miR-150, miR-425; D2 (ZT18–ZT24)—miR-148a, miR-148b, let-7g, miR-185; and L1 (ZT0–ZT6)—miR-30d. Results based on calculated acrophase are in agreement with findings based on D/L ratio, i.e., there are more miRNAs with peak in expression during D phase compared to L phase (Appendix A; Figure 4).

miRNAs selected for the analysis are highly conserved between rat and human (Appendix A), and rat miRNAs are not sufficiently annotated; therefore, human sequences were used to find miRNAs target genes in the database miRTarBase [87]. As rno-miR-7a-5p does not have a homologue in humans, this miRNA was not included in further steps of in silico analysis.

Functional analysis of target genes of miRNAs sorted according to the miRNA’s acrophase revealed that the proportion of genes influenced preferentially during the D phase is much higher compared to those that are influenced during the L phase of the LD cycle (Figure 6). A full list of target genes is provided in Appendix A.

Gene ontology enrichment analysis was performed with the use of the Panther database [88] and was focused on pathways influenced by target genes listed in Appendix A. Only those pathways that were influenced at least by three genes were included.

Genes interfering with miRNAs from D1 and D2 clusters influence apoptosis, angiogenesis, the immune system, and EGF and TGF-beta signaling to a higher extent compared to targets of L1 miRNA. Target genes of D1-clustered miRNAs influenced the Toll and Notch signaling pathways, which was not indicated for D2 and L1 miRNA targets. D2 miRNAs target genes were specific, in comparison with other clusters, to the regulation of hypoxia response via HIF activation.

The whole list of pathways cooperatively and/or selectively influenced by miRNA target genes is provided in Appendix A.

The genomic context of rhythmic miRNAs selected by screening in the study is provided in Appendix A and shows that most of the miRNAs are located in the intronic region of host genes and their seed sequence is broadly conserved among species, which implicates physiological impact.

The daily pattern in expression of miR-150-5p, miR-142-3p, and miR-30d-5p obtained by NGS was validated by real-time PCR (Figure 7A,D,G). miR-150-5p and miR-142-3p were selected because they exert the highest amplitude of daily rhythm with respect to the mean value from miRNAs with high expression exerting daily rhythm. To control the validity of the acrophase determined by NGS, miRNA with peak expression in antiphase was also included. Therefore, miR-30d-5p was selected for validation.

Cosinor analysis revealed that acrophase calculated based on NGS and PCR measurement are very similar. Peak expression of miR-150-5p was at ZT15.6 h (*p* < 0.05) measured by NGS and 16.6 h by PCR (*p* = 0.07) (Figure 7B). In the case of miR-142-5p, the maximum of expression was at ZT15.6 h (*p* < 0.05) according NGS and 15.7 h (*p* < 0.05) based on PCR results (Figure 7E). miR-30d-5p showed the highest expression at the transition from D to L phase of the LD cycle (ZT24.7 h and ZT22.8 h measured by NGS and PCR, respectively; *p* < 0.05, Figure 7H). Homogeneity of NGS and PCR measurement is also indicated by the significant positive correlation between data obtained by NGS and PCR (Figure 7C,F,I).

Expression of miR-150-5p exerted a significant daily rhythm in DLD1 cells. The rhythm in expression of the oncogene *myb* exerted an inverted acrophase compared to expression of miR-150-5p (cosinor, *p* < 0.05). The daily pattern in the expression of *bcl2* did not reach the level of significance in cosinor analysis (Figure 8A), although it was similar to that of *myb*. Transfection of miR-150-5p mimic caused an exponential increase in miR-150-5p concentration in DLD1 cells. Co-administration of miR-150-5p mimic with miR-150-5p inhibitor strongly reduced miR-150-5p increase measured after mimic transfection (Figure 8B). Administration of miR-150-5p mimic inhibited expression of *bcl2* and *myb* (Figure 8C and Figure 8D, respectively). Simultaneous exposure of cells to mimic and inhibitor prevented decrease in *bcl2* and *myb* expression induced by miR-150-5p mimic (Figure 8C and Figure 8D, respectively).

Screening of caspase 3 and/or 7 activation after miR-150-5p administration revealed potentiation of camptothecin effect by miR-150-5p. When DLD1 cells were exposed to camptothecin and miR-150-5p simultaneously, apoptotic cells were detected earlier in comparison to single-drug treatment. Therefore, we conclude that miR-150-5p promotes camptothecin-induced apoptosis (Figure 9A–D).

miR-150-5p mimic did not influence wound healing in DLD1 cell culture (Figure 9E); however, it significantly inhibited metabolism rate. Administration of miR-150-5p inhibitor increased metabolism rate in DLD1 cells (Figure 9F).

This study was also focused on the reciprocal influence of miR-150-5p on circadian oscillator functioning. We analyzed the effect of miR-150-5p manipulation on clock gene expression *cry1*, *cry2*, and *per2*. miR-150-5p mimic inhibited expression of *cry1*, and simultaneous administration of inhibitor weakened this influence from 28% to 18% inhibition (Figure 10A).

miR-150-5p administration was associated with increase in *per2* and *cry2* expression. This effect was eliminated by co-administration of miR-150-5p inhibitor (Figure 10B and Figure 10C, respectively).

Expression of miR-150-5p was significantly downregulated in human CRC tissue compared to adjacent tissue. This decrease was more pronounced in older patients (range 73–86 years, average 79.35 ± 3.89) compared to younger patients (range 37–68 years, 58.42 ± 8.46) (Figure 11A). The dependency of miR-150-5p expression in tumor on age was also demonstrated by the significant negative correlation between the age of patients and miR-150-5p expression in the tumor (y = −1.163x + 72.265, R = 0.301, *p* < 0.05 regression analysis) (Figure 11B).

Expression of *cry1* showed an inverted pattern with respect to miR-150-5p expression. *cry1* levels were increased in cancer compared to adjacent tissue, and this difference achieved the level of significance only in older patients (Figure 11C). A significant negative correlation was observed between miR-150-5p and *cry1* expression in older patients (y = −0.528x + 2703.9, R = 0.439, *p* < 0.05 regression analysis, Figure 11D). The clinical relevance of miR-150-5p dependency on age is demonstrated by Kaplan–Meier survival curve calculated specifically for younger (Figure 11E) and older patients (Figure 11F). While in younger patients, splitting of the cohort according to high and low miR-150-5p expression was not associated with 5-year survival (Figure 11E), we observed significantly better survival in older patients with high expression of miR-150-5p compared to those with low miR-150-5p expression (Figure 11F).

## 4. Discussion

This study was focused on the daily profile of miRNA expression in the rat colon with extrapolation of results with respect to the CRC context. We performed screening of mature and pre-miRNA expression under synchronized conditions to mimic the standard environmental regimen in humans. Screening revealed that nearly 10% of mature and 5% of precursor miRNAs exerted a rhythmic profile. Interestingly, in most cases expression achieved peak levels at the D1 phase of LD cycle, following by D2 and L1 phase. We did not detect any miRNA with acrophase within the L2 phase of the LD cycle. A daily pattern in miR-150-5p, miR-142-3p, and miR-30d-5p expression detected by NGS was validated in the rat colon by real-time PCR, and in the case of miR-150-5p, also in the CRC cell line DLD1.

Target genes that are regulated predominantly by D1- and D2-clustered miRNAs were associated with the regulation of apoptosis, EGF and TGF-beta signaling, the immune system, and angiogenesis. The role of miR-150-5p in apoptosis regulation was analyzed in DLD1 cells. miR-150-5p potentiated camptothecin-induced apoptosis and inhibited oncogenes involved in apoptosis regulation *myb*, *bcl2*, and *cry1* Expression of *myb* exerted a rhythmic profile in DLD1 cells with inverted acrophase with respect to miR-150-5p.

Our results are in accordance with the previous observations that miR-150 exerts a rhythmic pattern in the mouse liver [17,18] and human plasma [26]. In human plasma, a rhythmic pattern was also observed in levels of miR-139 [26]. let-7g, miR-30d, miR-148a, and miR-148b have been indicated as those miRNAs that are likely to be associated with the circadian rhythms based on the presence of putative sites for the circadian transcription factors in their promoters [24]. miR-142 showed a rhythmic pattern in a cell culture of NIH3T3 and 293ET cells following serum shock [16]. Therefore, there is a good agreement between the cycling miRNAs identified in the present study and independent resources. In accordance with previous observations [17], most of the rhythmic miRNAs are located in the introns of host genes (Appendix A).

The reason why the majority of miRNA in the rat colon exerts its maximum in the D phase of the LD cycle, which is an active phase in rats, is not known. The circadian system has a huge impact on the transcriptome [93]. Nearly one thousand genes with rhythmic expression have been identified in the gut of mouse during a 24 h cycle [94]. As miRNAs are frequently located within host genes, rhythms in miRNA levels could be produced coincidentally along with mRNA synthesis of their host genes. The gastrointestinal tract is being activated predominantly during the active phase of the LD cycle, which could lead to a higher proportion of transcriptional activity in the D phase in rats.

On the other hand, our result revealed low-amplitude rhythm in mRNA of DGCR8, which is an essential partner of Drosha in miRNA processing in the nucleus. We observed a pronounced trend to higher levels of DGCR8 mRNA during the D phase of the LD cycle compared to the L phase. A crucial role of DGCR8 in miRNA biosynthesis has been shown with the use of knockout experimental models [95]. Therefore, we suppose that acrophase in mRNAs’ daily profile can also be influenced by changes in DGCR8 expression. In rats, a clear-cut rhythm in the expression of DGCR8 mRNA was also detected in the liver, kidney, and heart [21]. However, other regulatory influences including transcription factors, long non-coding RNAs, methylation, and even other miRNAs [96,97] cannot be excluded.

In the present study, we observed more mature sequences that exert a daily rhythm in expression than their precursors. This could be caused by several reasons: in some cases, both strands of miRNA exert a rhythmic profile, and to the contrary, in some cases there was more than one coding gene for miRNA with arhythmic expression. miRNAs with very low expression could cause discrepancies.

The ratio of miRNAs expression in D vs. L phase showed a negative relationship with the corresponding miRNA abundance. This observation indicates that slower miRNA turnover can contribute to a flattening of miRNAs’ rhythmic profile [9].

According to GO enrichment analysis, target genes of miRNAs exerting acrophase in the D1, D2, and L1 clusters are involved in the regulation p53 pathway (P00059) and Wnt signaling (P00057), which is in accordance with previous findings [24]. However, in most cases, overlapping molecular pathways attributed to miRNAs from different clusters are executed via inhibition of non-identical target genes. While miR-128, miR-129, miR-139, miR-150, and miR-425 in the D1 miRNA cluster regulate Wnt signaling via CTNNB1, APC, and WNT4, let-7g, miR-148a, and miR-148b in the D2 cluster exert their effect via WNT10B, WNT1, and MYC (miRTarBase).

Apoptosis regulation is strongly associated with miRNAs with their peak in the D phase of the LD cycle by influencing genes BCL2, CAS3, and TP53 associated with hsa-miR-139-5p, hsa-miR-148a-3p, hsa-let-7g-5p, and/or miR-150-5p. This finding is supported by external evidence (e.g., [98]; #MIRT732599, MirTarBase).

GO pathways that are exclusively influenced by target genes of miRNA from D clusters include EGF and TGR-beta signaling pathways (P00018 and P00052, respectively). Target genes belonging to these pathways include, for example, KRAS, EGFR, PTEN, PIK3CA, TGFBR1, and SMAD2 and are cumulatively associated with miRNAs let-7g, miR-128, miR-148a, hsa-miR-425, miR-139, miR-150, and miR-128-3p (miRTarBase). KRAS, EGFR, PIK3CA, TGFBR1, and SMAD2 are convincingly linked to CRC progression [99,100,101]. In this context, miRNAs inhibiting expression of PTEN (miR-425 and miR-148a) should be considered as oncomirs, and those inhibiting KRAS, EGFR, PIK3CA, TGFBR1, and SMAD2 (let-7g, miR-150, miR-128, miR-139, miR-148a, miR-148b) as tumor suppressors.

Target genes of D miRNA clusters also influence the immune system via chemokine and cytokine signaling (P00031, P00036), although mostly via different genes. While D2 miRNA targets exert their functions by inhibition of STAT3, IL13, and IL15, target genes associated with D1 miRNAs act via inhibition of BTK, NFKB1, CXCR4, and CCR6 (miRTarBase), which are cumulatively influenced by let-7g, miR-139, miR-148a, miR-150, and miR-425. The above-mentioned genes, with the exception of IL15, have been proven to promote CRC progression and possess oncogenic potential [76,102,103,104,105,106,107]. Therefore, in this respect, let-7g, miR-139, miR-148a, miR-150, and miR-425 seem to be tumor suppressors rather than oncomirs. Target genes of the D1 miRNA cluster are specific to the regulation of the Toll receptor signaling pathway (P00054) and Notch signaling pathway (P00045). Target genes associated with the D2 miRNAs influence hypoxia response via HIF activation (P00030).

Although according to GO enrichment analysis, there are no pathways that would be regulated exclusively by miR-30d, there are some genes that are implicated as target genes only for this miRNA, e.g., hsa-miR-30d was shown to inhibit epithelial-to-mesenchymal transition in several types of cancer cells by inhibition of E-cadherin-transcriptional repressor SNAIL1 [108,109]. Similarly, hsa-miR-30d inhibits Wnt signaling via interacting with BCL9 [110]. SNAIL1 and BCL9 are not targeted by genes associated with the D clusters of miRNAs. In this respect, miR-30d can be considered to be a tumor suppressor.

As according to the in silico analysis, apoptosis is to a high extent controlled by genes under the regulatory influence of miRNAs from the D1 cluster, we tested the effect of miR-150-5p in this context. miR-150-5p was selected as it exerts the highest amplitude and the rhythmic pattern in its expression was also shown in several other tissues [17,18,26]. The impact of miR-150-5p administration on wound healing and metabolism in DLD1 cells was evaluated as well. DLD1 is a near-diploid cell line possessing Y chromosome derived from the large intestine of an adult male patient diagnosed with poorly or moderately differentiated colon adenocarcinoma. DLD1 cells bear the pS241F mutation in the *tp53* gene on one allele, while the other allele is silent. pS241F mutation causes generation of non-functional p53 protein. The mutated form *tp53* is also present in up to 50% of CRC patients [111,112].

Although miR-150-5p did not exert an effect on wound healing, it significantly inhibited metabolism in DLD1 cells. miR-150-5p also potentiated camptothecin-induced apoptosis. Inhibitory effects of miR-150 on CRC progression were previously observed in cell lines SW480, HC116, and HCT8 [56,113,114]. Potentiation of camptothecin’s pro-apoptotic effect by miR-150-5p was not reported previously, but miR-150-5p was shown to influence the effect of paclitaxel and oxaliplatin [115,116].

Mechanisms involved in miR-150-5p-induced apoptosis may include *myb*, *bcl2*, and *cry1*. *myb* is a well-known oncogene with anti-apoptotic properties [117] targeted by miR-150-5p [53,55,118]. Rhythmicity in *myb* expression was not reported with respect to cancer progression before. Rhythmic pattern in *myb* expression was detected in the lungs [119] and pituitary [120]. On the other hand, in datasets of Zhang et al. [5] and Mure et al. [121] *myb* was not listed among cycling genes.

*bcl2* is a pro-survival, anti-apoptotic protein [122] induced by *myb* [123,124]. Downregulation of *bcl2* by miR-150-5p has been previously reported in colonic epithelial cells [55]. An oncogenic role via anti-apoptotic effect has also been attributed to the clock gene *cry1* [68,125,126]. Its tumor-promoting role in CRC progression was demonstrated by *cry1*-induced cell proliferation and inhibited apoptosis in HT29 and SW480 CRC cell lines [68].

A tumor-suppressor role is usually attributed to miR-150 in CRC [53,114], which is more often downregulated in CRC tumor than the opposite [127,128,129,130]. The oncostatic capacity of miR-150-5p was demonstrated by observation that miR-150-5p delivery in exosomes suppressed migration and viability in CRC cell model SW620 [130]. Administration of miR-150-5p mimic induced apoptosis and inhibits metabolism of LoVo cells. miR-150-5p also inhibited tumor growth and induced intra-tumor apoptosis in mice bearing LoVo xenografts [53]. Moreover, low expression of miR-150-5p is associated with worse survival compared to high expression [128,130].

miR-150 also acts as a tumor suppressor in lymphoma [131] and esophageal cancer [132,133]. It plays a protective role in lung cancer [134,135], and a tumor suppressor role of miR-150 was confirmed in liver cancer [136,137]. On the other hand, the anti-apoptotic effect of miR-150 was implicated in gastric cancer [138,139], and an oncogenic role was attributed to miR-150-5p also in breast cancer [140]. The oncogenic effect of miR-150 in breast and gastric cancer has been questioned in following studies (e.g., [141] and [142], respectively). Therefore, tumor-suppressive effects of miR-150-5p seem to be generally prevailing.

Similarly, miR-142-3p and miR-30d-5p, whose rhythmic pattern was validated by real-time PCR, are more often implicated as tumor suppressors than the opposite. Downregulation of miR-30d has been detected in colorectal tumor tissues more often than upregulation [143,144,145,146,147]. Overexpression of miR-30d showed tumor suppressive effects in both in vitro and in vivo experiments. *LRH1* [148], *ATG5*, *PIK3*, *Beclin1* [149], and *GNA13* [150] have been proven to be direct targets of miR-30d. Therefore, it is assumed that miR-30d acts as a tumor suppressor in colorectal cancer.

miR-30d exerts tumor-suppressive effects in non-small-cell lung cancer [151,152,153,154,155,156]. In pancreatic cancer [157,158,159] and gallbladder cancer [160,161], a protective role of miR-30d is proposed. An oncostatic effect of miR-30d has been shown in prostate cancer cell lines [162,163], but an older study showed an oncogenic effect under in vitro and in vivo conditions [164]. Further research is needed to specify miR-30d’s effect in breast cancer [165,166].

Up- as well as downregulated miR-142-3p expression was reported in cancer compared to adjacent tissue in CRC patients [167]. A tumor-suppressive role of miR-142-3p was shown in CRC. Upregulation of miR-142-3p suppressed growth of CRC cells and xenograft tumor by downregulating *CD133*, *ABCG2*, and *Lgr5* [168], *CDK4* [169], and *CTNNB1* [170]. One study reports miR-142-3p stimulating migration and invasion of CRC cells [171]. There is also evidence that inhibition of the miR-142-3p/miR-506-3p-TGF-β1 axis by circPACRGL sponging of miRNAs promotes CRC cell proliferation, migration, invasion, and neutrophils differentiation [172].

miR-142-3p shows tumor-suppressive effects in hepatocellular cancer [173,174,175,176,177,178,179], ovarian cancer [80,180,181], and gastric cancer [141,182]. On the other hand, an oncogenic role of miR-142-3p was shown in gastric cancer [183,184], renal cancer [185,186], and prostate cancer [187].

The circadian system is a potent regulator of gene expression, including those coding miRNAs. It has been shown that miR-142-3p is in NIH3T3 cells under circadian control mediated by E-box [16], which is also indicated by our study. The daily pattern in miR-150-5p expression can be generated indirectly by *c-myc*, which is a clock-controlled gene [188] showing pronounced daily rhythm in all 64 tissues examined in baboon [121]. Expression of miR-150-5p is progressively inhibited by *c-myc* [189,190]. The assumption that *c-myc* generates the rhythm in miR-150-5p expression is in accordance with phase shift between miR-150-5p and *c-myc* acrophases. Rhythm in miR-30d-5p expression has not been reported previously to our knowledge. However, it has been shown that 126 miRNAs, including miR-30d-5p, are regulated by glucose levels [191]. Access to food is a potent synchronizer of mRNA expression in the liver and gastrointestinal tract [192,193,194]. Considering the results of Tang et al. [191], food and its composition are likely candidates for synchronization of miR-30d expression.

A more or less complex regulatory feedback loop between clock genes and miRNAs has been described previously, e.g., *per2* vs. miR-34a-5p [195] or *bmal1* vs. miR-142-3p [16]. Here, we report that miR-150-5p influences the intracellular circadian oscillator by targeting clock gene *cry1*. The physiological relevance of this observation has been validated with the use of a human cohort. We observed an age-dependent decrease in miR-150-5p expression in human CRC tissue compared to adjacent tissue, with a more pronounced decrease in miR-150-5p expression in older patients. This decrease was accompanied by an increase in the expression of clock gene *cry1* in tumor compared to adjacent tissue. In vitro transfection of miR-150-5p confirmed that *cry1* is a target gene of miR-150-5p. Older patients with lower expression of miR-150-5p and higher expression of *cry1* showed worse survival in comparison with those with low miR-150-5p expression.

Age-dependent changes in miR-150-5p have not been reported previously. However, it was shown that lncRNA ZFAS1 targeting miR-150-5p exerts age-dependent expression in hepatocellular carcinoma, with higher expression in older compared to younger patients. Therefore, sponging by ZFAS1 can drive an age-dependent pattern in miR-150 levels. Changes in miR-150 availability can influence *cry1* expression with higher intensity in older patients compared to younger ones [196]. Age-dependent expression of *cry1* in tumors of CRC patients has been previously reported by Mazzoccoli et al. [68].

### Limitations of the Study

Study limitations include the utilization of very strict statistical analysis that was used to omit false positive results. Rhythmic miRNAs with low amplitude could escape in this way from screening. In our in silico analysis, we did not take into consideration miRNAs with median and low expression, as miRNAs with high concentration are more likely to cause some physiological effect. However, low-abundance miRNAs can exert a regulatory role at the single-cell level within some specific biological contexts.

This study does not include in vivo functional validation of miR-150-5p’s effect on *myb*, *bcl2*, and *cry1* expression or tumor growth. A role of miR-150-5p/*myb* interference in CRC tumor growth has been demonstrated previously in nude mice [53]; however, the tumor suppressor role of miR-150-5p mediated via *bcl2* and *cry1* expression needs to be addressed in the next studies.

The present study does not include evidence that miR-150-5p inhibits protein products of *bcl2*, *myb*, and *cry1*. However, an inhibitory effect of miR-150-5p on *bcl2* and *myb* translation was already demonstrated in CRC cell lines LoVo and HT27 elsewhere [53,55]. A high level of correlation between mRNA and protein products biosynthesis of clock gene [197,198,199] and *cry1* in particular was also shown [200]. Therefore, we suppose that the mRNA-based evidence provided in our study is satisfactory for conclusions.

As CRC patients frequently bear *tp53* mutation [112], the DLD1 cell line with mutated *tp53* was used to evaluate results of in silico analysis experimentally. It is possible that miR-150-5p effects would differ in other *tp53* contexts. On the other hand, the majority of independent experimental studies [53,54,55,56,113,114,115,116] also indicate a tumor suppressor function of miR-150-5p.

## 5. Conclusions

CRC belongs among the most frequent malignancies and is strongly associated with disturbances in miRNA expression. miRNAs are being suggested as CRC markers and are being tested for the treatment of solid cancer in translational medicine [201]. Unfortunately, little is known about daily patterns in miRNA expression in colon tissue. Therefore, the aim of present study was to analyze mature and premature miRNA expression in the colon during a 24 h cycle and perform ontological analysis of genes associated with rhythmic miRNAs.

Approximately 10% of abundantly expressed miRNAs in the colon exerted a distinct daily rhythm. Peak expression of rhythmic miRNAs was, in the vast majority of cases, observed during the D phase of the LD cycle, which corresponds with the pattern in DGCR8 mRNA expression and the active phase of the LD cycle of the rat, when digestion and metabolism arise.

Rhythmic miRNAs were assigned to clusters according to acrophase. Apoptosis was strongly influenced by miRNAs with peak during the D phase of the LD cycle. Effects of miR-150-5p were tested in this respect. Administration of miR-150-5p inhibited metabolism of DLD1 cells and potentiated camptothecin-induced apoptosis. miR-150-5p can facilitate apoptosis by targeting oncogenes *bcl2* and *myb* or clock gene *cry1*. *myb* exerted a rhythmic pattern in expression in DLD1 cells, with peak value in antiphase with respect to miR-150-5p expression. Age-dependent miR-150-5p expression in human CRC tissue was reflected by the inverted pattern of *cry1*. Older patients with lower expression of miR-150-5p and higher expression of *cry1* showed worse survival in comparison with younger patients.

Results indicate that the timing of miRNA expression, e.g., miR-150-5p targeting *myc*, *bcl2* and *cry1*, can influence specific molecular pathways, including apoptosis involved in the regulation of CRC progression.

## Figures and Tables

**Figure 1 biomedicines-13-01865-f001:**
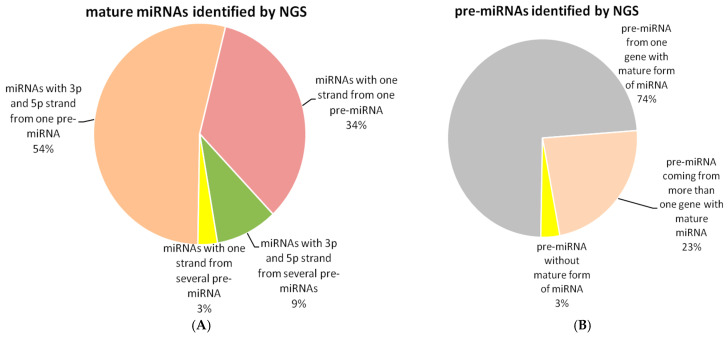
Overview of mature and premature forms of miRNAs and coding genes identified by NGS screening: (**A**) mature miRNAs sorted according to expression of 3p and 5p strands and pre-miRNA; (**B**) pre-miRNAs sorted according to expression of mature form and number of coding genes. Majority of NGS-identified mature miRNAs express both strands (5p and 3p) and are synthesized from one pre-miRNA. The majority of pre-miRNA is coded by one gene.

**Figure 2 biomedicines-13-01865-f002:**
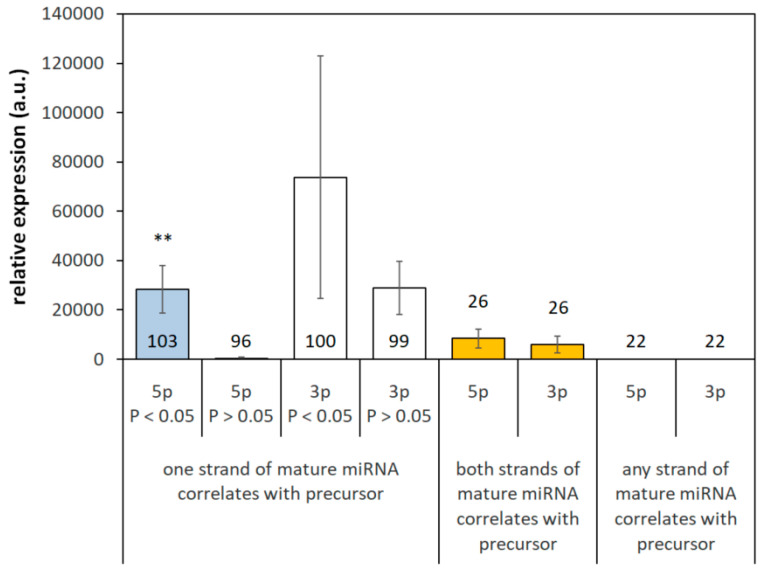
Correlation between intensity of miRNA and pre-miRNA expression coded by one gene. When mature miRNA was generated from one gene and expression of pre-miRNA correlated only with one strand of mature miRNA, there was not a significant difference between 5p (blue columns) and 3p (white columns) strands. However, in the case of 5p strands, expression of correlating strands was significantly higher compared to non-correlating 5p sequences. In the case of 3p strands, higher expression in correlating 3p sequences compared to those that do not correlate with pre-miRNA was observed only as a trend. *p* < 0.05—indicated strand significantly correlates with pre-miRNA; *p* > 0.05—indicated strand does not correlate with pre-miRNA, ** *p* < 0.01, comparison of correlating and non-correlating 5p strands with respect to precursor miRNA, unpaired *t*-test. There is a difference in how the 5p and 3p strand is generated from pre-miRNA.

**Figure 3 biomedicines-13-01865-f003:**
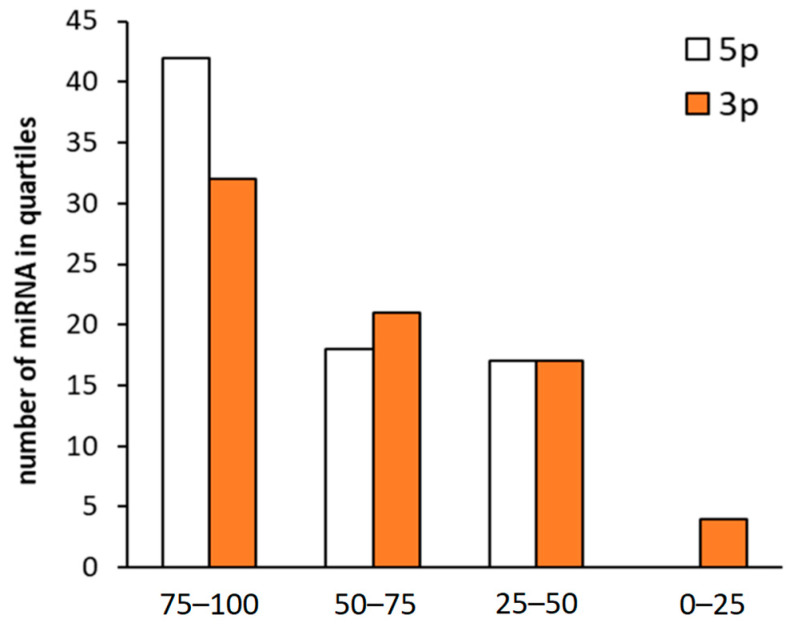
Distribution of 5p and 3p strands derived only from one gene whose expression correlates with pre-miRNA sorted into four quartiles according to the expression intensity. Positive correlation between mature miRNA and pre-miRNA expression is associated with high expression of mature miRNA.

**Figure 4 biomedicines-13-01865-f004:**
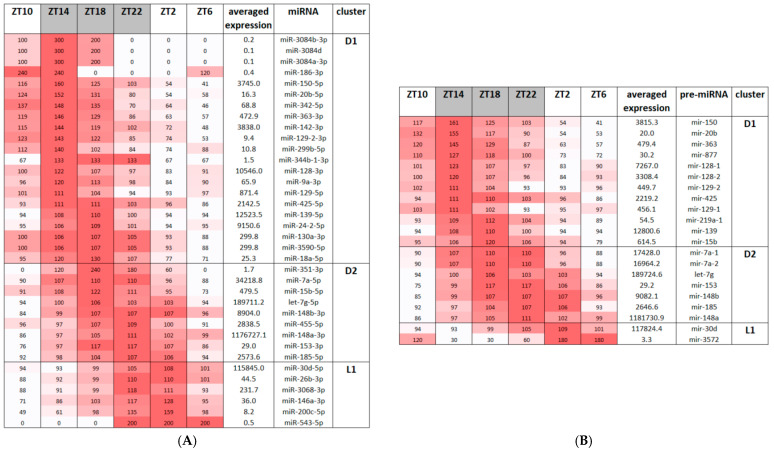
Heat map of miRNA (**A**) and pre-miRNA (**B**) expression. L—light, D—dark phase of 24 h LD cycle; D1—maximum expression during the first half of D phase of LD cycle, D2—maximum expression during the second half of D phase of LD cycle; L1—maximum expression during the first half of L phase of LD cycle. ZT—relative time units, Zeitgeber time. Beginning of L phase is defined as Zeitgeber time 0 (ZT0). Beginning of D phase is defined as Zeitgeber time 12 (ZT12). D phase is indicated by grey color in the first row of both tables. Scale of red color indicates intensity of expression. Deep red color indicates highest expression, white color indicates lowest expression. Most mature and pre-miRNAs exert their maximum in rhythmic expression in the D phase of the LD cycle.

**Figure 5 biomedicines-13-01865-f005:**
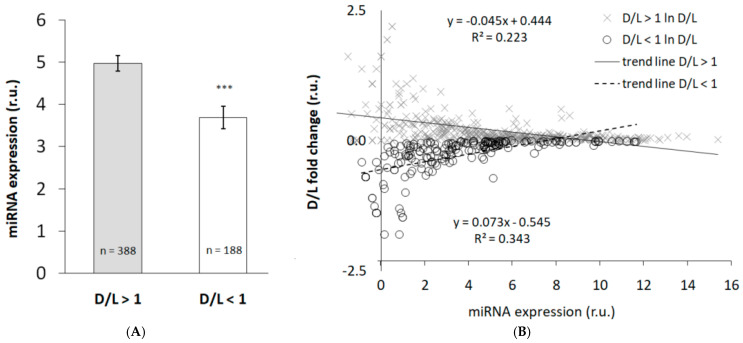
Expression of miRNAs relativized according to the phase of LD cycle and its relationship to their abundance: (**A**) ratio of miRNA expression measured during the dark (D) compared to light (L) phase split according to intensity of expression in D, *** *p* < 0.001, comparison of D/L > 1 vs. D/L < 1, unpaired *t*-test; (**B**) relationship between D/L ratio and intensity of expression. As distribution of miRNA expression did not show normal distribution, data are presented after log normalization. r.u.—relative units. Intensity of miRNA expression shows negative correlation with amplitude of daily profile in miRNA expression.

**Figure 6 biomedicines-13-01865-f006:**
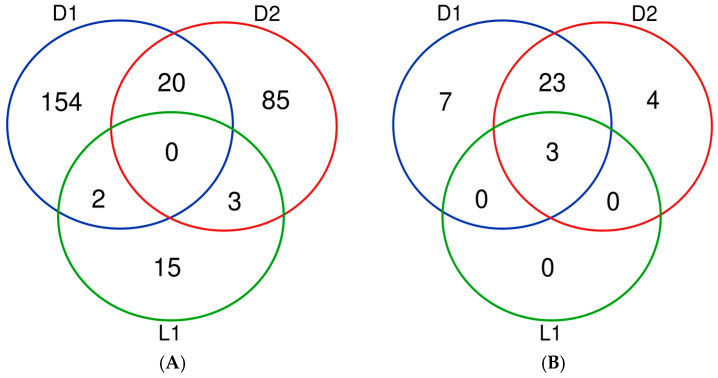
Venn diagram of (**A**) number of target genes of miRNAs sorted according to their acrophase; (**B**) number of regulatory pathways influenced by target genes. D1—the first half of the D time of 24 h cycle (ZT12–ZT18); D2—the second half of the D time of 24 h cycle (ZT18–ZT24); L1—the first half of the L time of 24 h cycle (ZT0–ZT6). Most genes are targeted by rhythmic miRNA belonging to D1 phase of LD cycle. miRNAs with peak expression in D1 and D2 phases show a significant overlap in pathways activated by their target genes. This is not observed with respect to miRNAs from the L phase of the LD cycle.

**Figure 7 biomedicines-13-01865-f007:**
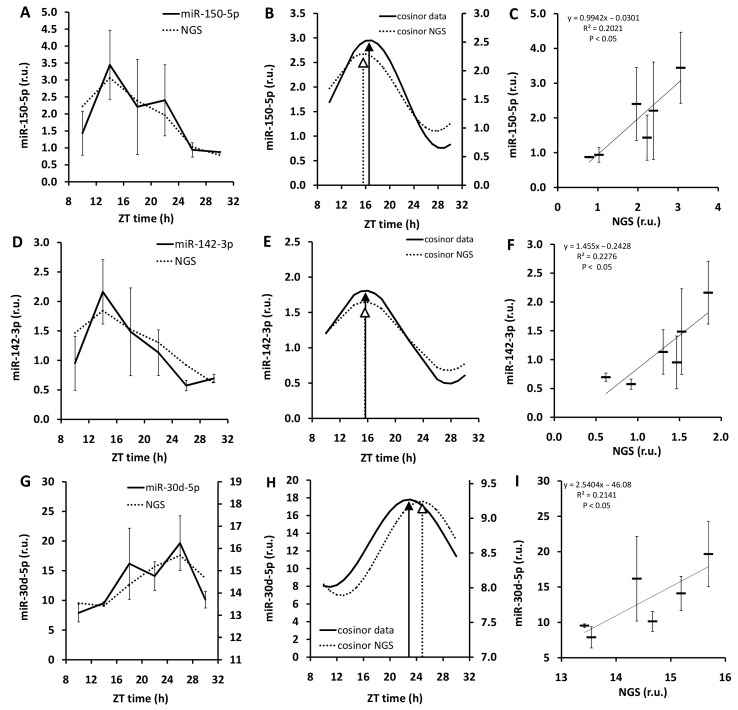
Rhythmic expression of miRNAs in the rat colon measured by real-time PCR (PCR) and its correlation with results of NGS. Daily pattern in expression of miR-150-5p (**A**), miR-142-3p (**D**), and miR-30d-5p (**G**) measured by PCR (solid line) and NGS (dotted line). Data are provided as mean ± SEM. (**B**,**E**,**H**) display the best cosinor fit calculated based on PCR (solid line) and NGS (dotted line) measurement. Black arrow indicates acrophase according to PCR; white arrow shows acrophase according to NGS. (**C**,**F**,**I**) demonstrate correlation between the PCR and NGS results. Data are provided as mean (thick horsontal line) ± SEM (vertical line). NGS—next generation sequencing, ZT—Zeitgeber time, h—hours, r.u.—relative units. NGS-detected daily rhythm in miR-150-5p, miR-142-3p, and miR-30d-5p expression was validated by PCR.

**Figure 8 biomedicines-13-01865-f008:**
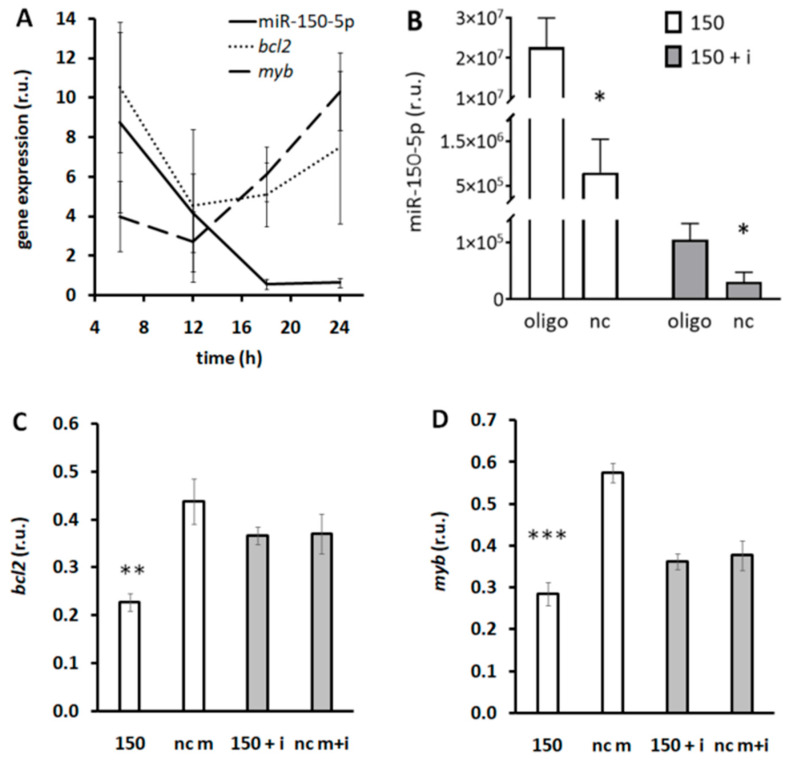
Daily pattern in expression of miR-150-5p and its target genes in DLD1 cells: (**A**) expression of miR-150-5p, *bcl2*, and *myb* during 24 h cycle in DLD1 cells synchronized by serum shock; (**B**) efficiency of transfection by miR-150-5p mimic or mimic administered together with inhibitor in DLD1 cells. oligo—cells transfected with miR-150-5p inhibitor and/or mimic, nc—cells transfected with control sequence(s). Cells transfected with experimental oligos were compared to nc, * *p* < 0.05—unpaired *t*-test. Effect of miR-150-5p, miR-150-5p inhibitor, and control oligos on *bcl2* (**C**) and *myb* (**D**) expression in DLD1 cells. 150—miR-150-5p mimic, 150 + i—miR-150-5p mimic administered together with inhibitor; nc m—control sequence for mimic, nc m + i—control sequences for mimic and inhibitor administered together. Expression in cells transfected with miR-150-5p was compared to nc m; expression of cells transfected with mimic together with inhibitor was compared to nc m + i. ** *p* < 0.01, *** *p* < 0.001—unpaired *t*-test, r.u.—relative units. Expression of miR-150-5p and *myb* exerts a rhythmic patter with peak levels in antiphase in DLD1 cells. miR-150-5p mimic administration inhibits expression of *bcl2* and *myb*.

**Figure 9 biomedicines-13-01865-f009:**
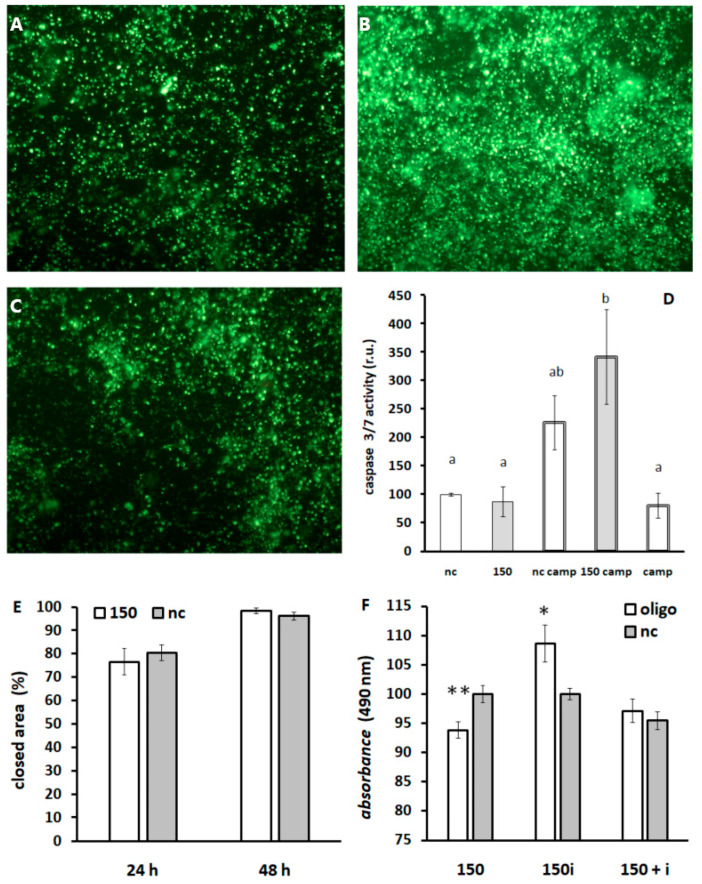
Effect of miR-150-5p mimic (150) and/or camptothecin (camp) on apoptosis and effect of miR-150-5p on wound healing and metabolism was tested in DLD1 cells. Representative images of cells with activated caspase 3 and/or 7 show effect of (**A**) control oligos (nc), (**B**) miR-150-5p mimic and camp together, and (**C**) camp alone. Images were taken with an inverted fluorescence microscope with a magnification of 100×. (**D**) summary of fluorescent staining showing apoptotic intensity after miR-150-5p mimic and/or camp administration. (**E**) effect of miR-150-5p on wound healing in DLD1 cell culture; (**F**) effect of miR-150-5p mimic, miR-150-5p inhibitor (150i), and miR-150-5p mimic together with inhibitor (150 + i) on metabolic activity of DLD1 cells. Histograms show mean ± SEM. Different letters above columns indicate significant differences between groups at the level of *p* < 0.05 calculated by ANOVA followed by Tukey’s post hoc test (**D**). Comparison between experimental oligos and corresponding negative control (**E**,**F**); * *p* < 0.05, ** *p* < 0.01—unpaired *t*-test. miR-150-5p facilitated early phase of camptothecin-induced apoptosis. Ectopic miR-150-5p decreased metabolic rate of DLD1 cells, and administration of inhibitor together with miR-150-5p mimic inverted this effect.

**Figure 10 biomedicines-13-01865-f010:**
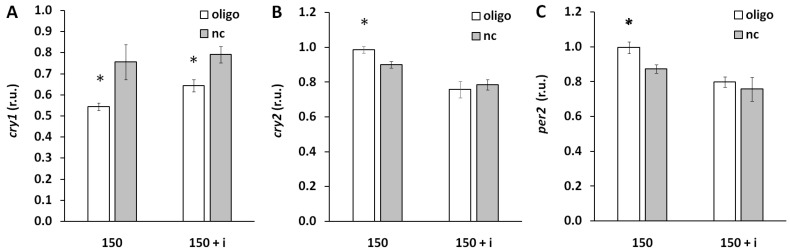
Effect of miR-150-5p administration on expression of clock genes *cry1* (**A**), *cry2* (**B**), and *per2* (**C**) in DLD1 cells. Cells were transfected with miR-150-5p (150) or miR-150-5p together with inhibitor (150 + i). Effect of miR-150-5p mimic or mimic administered together with inhibitor (oligo) was evaluated with respect to cells transfected with corresponding control sequences (nc). Comparison between experimental oligos and corresponding negative control; * *p* < 0.05—unpaired *t*-test, r.u.—relative units. miR-150-5p administration inhibited expression of *cry1* and induced expression of *cry2* and *per2*.

**Figure 11 biomedicines-13-01865-f011:**
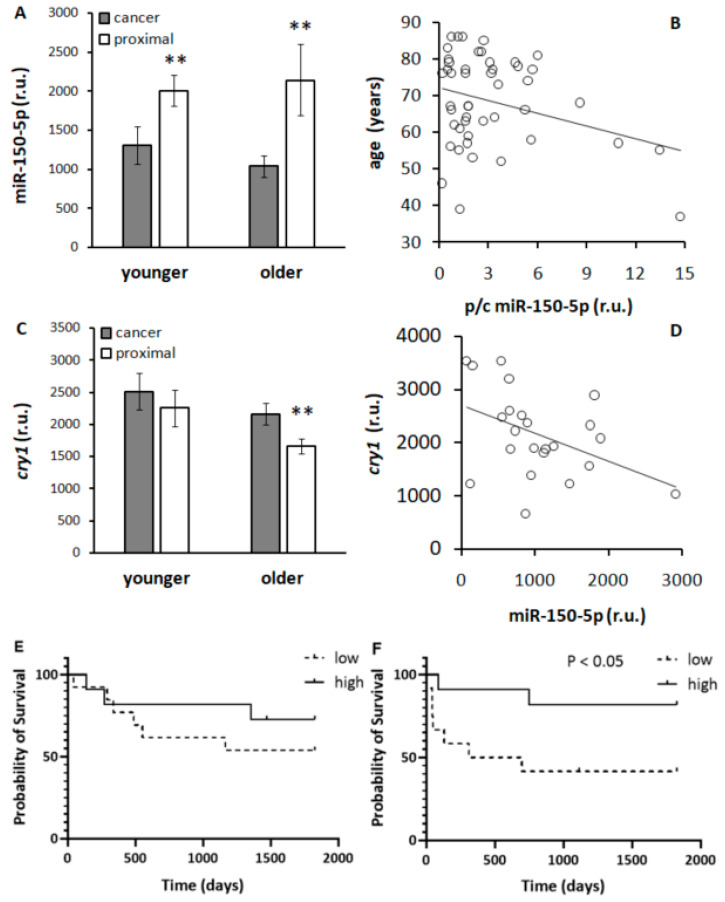
Expression of miR-150-5p in tumor and adjacent proximal tissue in CRC patients negatively correlates with age: (**A**) expression of miR-150-5p in tumor and adjacent proximal tissue in younger (≤median, range 37–68 years (58.42 ± 8.46) and older (>median, range 73–86, 79.35 ± 3.89, years) CRC patients; (**B**) negative correlation between age and ratio of miR-150-5p expression in proximal vs. cancer tissue, regression is indicated by solid line; (**C**) *cry1* expression in tumor and adjacent tissue of CRC patients sorted according to age; (**D**) negative correlation between miR-150-5p and *cry1* mRNA expression in older CRC patients, regression is indicated by solid line; (**E**) Kaplan–Meier survival curve showing 5-year survival of younger and (**F**) older CRC patients. low—miR-150-5p expression ≤ median (dotted line), high—miR-150-5p expression > median (solid line) in tumor tissue. *p* < 0.05 level of significance in log-rank test (**F**); comparison between cancer vs. proximal tissue, ** *p* < 0.01—unpaired *t*-test (**A**,**C**), r.u.—relative units. Expression of miR-150-5p is lower in cancer compared to adjacent tissue and this difference is more prominent in older compared younger CRC patients. miR-150-5p shows a negative correlation with *cry1* expression in older patients. Older patients with low miR-150-5p expression show worse survival compared to those with high miR-150-5p expression.

**Table 1 biomedicines-13-01865-t001:** miRNA with rhythmic profile sorted by quartiles reflecting expression intensity.

3.8%	2.2%	1.9%	1.9%
75–100	50–75	25–50	0–25
Very High Expression	High Expression	Low Expression	Very Low Expression
miR-30d-5p	miR-26b-3p	miR-18a-5p	miR-351-3p
miR-7a-5p	miR-363-3p	miR-20b-5p	miR-186-3p
miR-185-5p	miR-15b-5p	miR-153-3p	miR-543-5p
miR-425-5p	miR-3068-3p	miR-200c-5p	miR-344b-1-3p
miR-150-5p	miR-9a-3p	miR-146a-3p	miR-3084b-3p
miR-128-3p	miR-130a-3p	miR-299b-5p	miR-3084d
miR-455-5p	miR-3590-5p	miR-129-2-3p	miR-3084a-3p
miR-129-5p	miR-342-5p		
miR-139-5p			
miR-24-2-5p			
let-7g-5p			
miR-148b-3p			
miR-148a-3p			
miR-142-3p			

**Table 2 biomedicines-13-01865-t002:** pre-miRNA with rhythmic profile sorted by quartiles reflecting expression intensity.

2.2%	1.3%	0.9%	0.2%
75–100	50–75	25–50	0–25
Very High Expression	High Expression	Low Expression	Very Low Expression
mir-30d	mir-185	mir-20b	mir-3572
mir-7a-2	mir-425	mir-153	
mir-7a-1	mir-15b	mir-219a-1	
mir-150	mir-363	mir-877	
mir-128-1	mir-129-1		
mir-128-2	mir-129-2		
mir-139			
let-7g			
mir-148b			
mir-148a			

The first row shows the percentage of pre-miRNAs in a particular cluster with respect to all detected pre-miRNAs.

## Data Availability

All datasets and supporting information are available in the article and Appendix A.

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
