# Peer review of "Daily Profile of miRNAs in the Rat Colon and In Silico Analysis of Their Possible Relationship to Colorectal Cancer"

_biomedicines, 2025, doi:10.3390/biomedicines13081865_

Round 1
Reviewer 1 Report (Previous Reviewer 1)
Comments and Suggestions for Authors
The manuscript titled "Daily profile of miRNAs in the rat colon and in silico analysis of their possible relationship to colorectal cancer" presents an ambitious and comprehensive investigation of circadian regulation of miRNA expression and its connection to colorectal cancer (CRC). The study utilizes a robust combination of NGS, RT-PCR validation, in vitro assays, and in silico analysis to identify miRNAs with rhythmic expression patterns and their functional implications in CRC, particularly focusing on miR-150-5p.
The manuscript is thorough and the results are compelling. Below are some comments and suggestions for the authors to consider:
Major Comments
— Clarify rationale for miRNA selection. While miR-150-5p, miR-142-3p, and miR-30d-5p are validated and followed up experimentally, the rationale for prioritizing these miRNAs over others identified as rhythmic is not fully explained. Please provide clearer justification for their selection based on expression levels, relevance to CRC, or prior literature.
— Functional confirmation in vivo. While in vitro data support the pro-apoptotic role of miR-150-5p, the lack of in vivo functional validation (e.g., in a rat CRC model or xenograft) limits the translational strength of the findings. Please discuss this limitation more explicitly in the Discussion.
— Control normalization strategy. For qPCR normalization, multiple reference genes (e.g., rnu6-1, snord47, β-actin) are mentioned. Please justify and clarify how the most stable housekeeping gene was selected for each assay, as normalization affects interpretation of rhythmicity.
— miRNA-target interactions. Although targets of miR-150-5p are validated at the mRNA level, protein-level validation (e.g., Western blot of BCL2, MYB) is not included. Please clarify if this was attempted or provide rationale for focusing only on mRNA.
— Batch effects and biological replicates. The methodology would benefit from a clearer statement regarding how batch effects and biological variability were controlled for across NGS, RT-PCR, and in vitro experiments, especially given the time-series design.
Minor Comments
— Abstract language. The abstract is informative but overly dense. Consider simplifying and clearly separating background, methods, results, and conclusions.
— Typographical errors. There are several minor typographical errors throughout the manuscript (e.g., “camptotecin” instead of “camptothecin”, and inconsistent spacing around punctuation). Please revise carefully.
— Figure legends. Some figure legends, especially for Figures 7–11, could benefit from more concise and self-contained descriptions of what is being shown and what the main finding is.
— Statistical annotations. Ensure consistent reporting of statistical results (e.g., p-values, confidence intervals, and n numbers). In some cases, "P" is capitalized, while in others "p" is used.
— Abbreviations. Please ensure all abbreviations are defined upon first use (e.g., DLD1, RF-EMF, LD cycle, ZT time).
In summary, this is a valuable and well-designed study that contributes new insights into the chronobiology of miRNAs and their role in CRC. Addressing the comments above will strengthen the manuscript’s impact and clarity.
Author Response
Dear reviewer,
Thank you very much for your valuable time and opinion, we appreciate it very much.
All your suggestions and comments were implemented into the text, please see detailed response bellow.
Changes in the text are indicated by blue background of the text.
Sincerely,
Iveta Herichová on behalf of the all authors
Reviewer 1
Comments and Suggestions for Authors
The manuscript titled "Daily profile of miRNAs in the rat colon and in silico analysis of their possible relationship to colorectal cancer" presents an ambitious and comprehensive investigation of circadian regulation of miRNA expression and its connection to colorectal cancer (CRC). The study utilizes a robust combination of NGS, RT-PCR validation, in vitro assays, and in silico analysis to identify miRNAs with rhythmic expression patterns and their functional implications in CRC, particularly focusing on miR-150-5p.
The manuscript is thorough and the results are compelling. Below are some comments and suggestions for the authors to consider:
Major Comments
— Clarify rationale for miRNA selection.
While miR-150-5p, miR-142-3p, and miR-30d-5p are validated and followed up experimentally, the rationale for prioritizing these miRNAs over others identified as rhythmic is not fully explained. Please provide clearer justification for their selection based on expression levels, relevance to CRC, or prior literature.
- thank you for the comment. miR-150-5p and miR-142-3p were selected because they exerted highest amplitude of daily rhythm in respect to mean value. Please, see table below. miR-30d-5p was selected to measure some miRNA exerting peak in antiphase and in this respect miR-30d-5p was the only possibility.
Explanation was incorporated into the chapter “Results”:
miR-150-5p and miR-142-3p were selected because they exert highest amplitude of daily rhythm in respect to mean value from miRNAs with high expression exerting daily rhythm. To control validity of acrophase determined by NGS miRNA with peak expression in antiphase was also included. Therefore, miR-30d-5p was selected for validation.
— Functional confirmation in vivo.
While in vitro data support the pro-apoptotic role of miR-150-5p, the lack of in vivo functional validation (e.g., in a rat CRC model or xenograft) limits the translational strength of the findings. Please discuss this limitation more explicitly in the Discussion.
- the lack of in vivo functional validation was discussed in study limitations in the chapter “Discussion”:
“Study does not include in vivo functional validation of miR-150-5p effect on myb, bcl2 and cry1 expression and tumour growth. A role of miR-150-5p/myb interference in CRC tumour growth has been demonstrated previously in nude mice [53]; however, tumour suppressor role of miR-150-5p mediated via bcl2 and cry1 expression needs to be addressed in next studies”.
— Control normalization strategy.
For qPCR normalization, multiple reference genes (e.g., rnu6-1, snord47, β-actin) are mentioned. Please justify and clarify how the most stable housekeeping gene was selected for each assay, as normalization affects interpretation of rhythmicity.
In real-time PCR we perform arbitrary quantification with standard curve specific for each particular tissue and gene (including housekeepers). Before the use of gene for normalisation we check sensitivity of housekeeper expression to treatment (and/or rhythmicity). This control is facilitated by StepOnePlus™ software and chemistry which allow use of ROX fluorescent dye for normalization of fluorescent signal for different plate wells to reduce eventual heterogeneity of measurement. After quantification of housekeeper expression with the use of calibration curve and appropriate statistical analysis we can detect if housekeeper expression (specifically for each tissue and gene) changes in respect to treatment and/or LD cycle. Only housekeeper that does not respond to treatment is used for normalisation.
This statement was included in the chapter “Materials and Methods”.
— miRNA-target interactions.
Although targets of miR-150-5p are validated at the mRNA level, protein-level validation (e.g., Western blot of BCL2, MYB) is not included. Please clarify if this was attempted or provide rationale for focusing only on mRNA.
- thank you for the comment. We agree that inhibitory influence of miRNA always issue into decrease of translation. The first studies really indicated that in miRNAs inhibit gene expression mainly by influencing translation. However, since that more complex miRNA mediated regulation of transcriptome was revealed and mRNA degradation in response to miRNA treatment was firmly proved (e.g. Guo et al., 2010, 3121 references in WOS). Later dominant role of mRNA degradation in execution of miRNA mediated effects was described in detail (e.g. Huntzinger and Izaurralde, 2011, 1792 references in WOS). Recently it is well accepted that miRNA mediated inhibition of proteosynthesis includes deadenylation and decaping of mRNA and consequent degradation of target mRNA (e.g. O'Brien et al., 2018, 2192 references in WOS).
Guo H, Ingolia NT, Weissman JS, Bartel DP. Mammalian microRNAs predominantly act to decrease target mRNA levels. Nature. 2010; 466(7308):835-40. doi: 10.1038/nature09267.
https://pubmed.ncbi.nlm.nih.gov/20703300/
Huntzinger E, Izaurralde E. Gene silencing by microRNAs: contributions of translational repression and mRNA decay. Nat Rev Genet. 2011; 12(2):99-110. doi: 10.1038/nrg2936.
https://pubmed.ncbi.nlm.nih.gov/21245828/
O'Brien J, Hayder H, Zayed Y, Peng C. Overview of MicroRNA Biogenesis, Mechanisms of Actions, and Circulation. Front Endocrinol (Lausanne). 2018; 9:402. doi: 10.3389/fendo.2018.00402.
https://pubmed.ncbi.nlm.nih.gov/30123182/
Based on finding that miRNAs influence mRNA levels “The experimentally validated microRNA-target interactions database” (miRTarBase) accepts qPCR as strong evidence of miRNA and target gene interaction.
https://mirtarbase.cuhk.edu.cn/~miRTarBase/miRTarBase_2022/php/index.php
e.g. interaction miR-150-5p – myb
miRTarBase is highly reputable database as it is indicated by more than 1600 references in WOS (sum for updates listed below):
Cui et al.: Nucleic Acids Res. 2025 53(D1):D147-D156
Huang et al.: Nucleic Acids Research 2022, 50(D1):D222-D230
Huang et al.: Nucleic Acids Research 2020, 48(D1):D148-D154
Regulation of clock genes occurs mainly at the transcriptional level via E-box in their promoter. Basic feedback loop controls transcription of so called clock controlled genes also via E-box (Reppert and Weaver, 2001). It is known that pronounced changes in clock gene expression are followed by change in corresponding protein level and therefore results based on measurement of clock gene mRNA expression are well accepted in biological rhythm oriented research (e.g. Yan et al., 1999; Oishi et al., 1998).
Reppert SM, Weaver DR. Annu Rev Physiol. 2001; 63:647-76. - 1284 citations in Scopus
Yan L, Takekida S, Shigeyoshi Y, Okamura H. Neuroscience. 1999; 94(1):141-50. - 219 citations in Scopus
Oishi K, Sakamoto K, Okada T, Nagase T, Ishida N. Biochem Biophys Res Commun. 1998; 253(2):199-203. - 204 citations in WOS
We included following statement in the section “Limitations of the study”:
Recent study does not include evidence that miR-150-5p inhibits protein products of bcl2, myb and cry1. However, inhibitory effect miR-150-5p on bcl2 and myb translation was already demonstrated in CRC cell lines LoVo and HT27 elsewhere [53,55]. High level of correlation between mRNA and protein products biosynthesis of clock gene [197,198,199] and cry1 in particular was also shown [200]. Therefore, we suppose that mRNA based evidence provided in our study is satisfactory for conclusions.
- Oishi, K.; Sakamoto, K.; Okada, T.; Nagase, T.; Ishida, N. Antiphase circadian expression between BMAL1 and period homologue mRNA in the suprachiasmatic nucleus and peripheral tissues of rats. Biochem Biophys Res Commun.1998, 253, 199-203, doi: 10.1006/bbrc.1998.9779.
- Yan, L.; Takekida, S.; Shigeyoshi, Y.; Okamura, H. Per1 and Per2 gene expression in the rat suprachiasmatic nucleus: circadian profile and the compartment-specific response to light. Neuroscience1999, 94, 141-150, doi: 10.1016/s0306-4522(99)00223-7.
- Reppert, S.M.; Weaver, D.R. Molecular analysis of mammalian circadian rhythms. Annu Rev Physiol.2001, 63, 647-676, doi: 10.1146/annurev.physiol.63.1.647.
- van der Watt, P.J.; Roden, L.C.; Davis, K.T.; Parker, M.I.; Leaner, V.D. Circadian Oscillations Persist in Cervical and Esophageal Cancer Cells Displaying Decreased Expression of Tumor-Suppressing Circadian Clock Genes. Mol Cancer Res.2020, 18, 1340-1353, doi: 10.1158/1541-7786.MCR-19-1074.
— Batch effects and biological replicates.
The methodology would benefit from a clearer statement regarding how batch effects and biological variability were controlled for across NGS, RT-PCR, and in vitro experiments, especially given the time-series design.
Elimination of batch effect in laboratory analysis:
Batch effects are excluded at all levels of laboratory measurement. During extraction samples selected for one set are mixed, to include samples from different time-points, different patients, different tissues (cancer vs. control) etc. Reverse transcription (RT) is organised in the same way, but the set of samples for RT always differ from that one used during isolation. In real-time PCR we always measure samples to be compared with the same master mix.
Elimination of batch during sampling:
In the time-series experiment we always measure expression of clock genes that are involved in basic feed-back loop in addition to those that are in the focus of the article. As clock genes per2 and bmal1 exerts inverted acrophase in their daily pattern of expression, we use them as an internal control generally (including the batch effect). It is assumed that as long as expression of per2 and bmal1 expression is in antiphase, experimental design was not influenced by batch effect.
There are two figures in the main MS showing results from the time-series experiments – Figure 7A, D and G (expression of miRNAs in the colon) and Figure 8A (expression of cry1, myb and bcl2 in DLD1 cells).
Expression of clock genes per2, bmal1 and rev-erbα is displayed in the supplementary Figure S3 included in the MS. Figure S3 demonstrates expected phase shifts between acrophases of rhythmic clock gene expression.
Figure S3
Daily pattern of per2, bmal1 and rev-erbα expression in the rat colon (A)
In DLD1 cells (Figure 8A) we also measured expression of per2 and bmal1 in the same samples like cry1, myb and bcl2. Expression of per2 and bmal1 at ZT6 and ZT18 was in antiphase as expected.
To conclude, batch effect did not influence time-series experiments in our study. Number of biological replicates was included in the chapter “Material and Methods”.
Following statement was incorporated in the chapter “Material and Methods” part “2.7. Statistical Analysis”.
Sample laboratory processing was randomised. During extraction samples selected for one set were mixed, to include samples from different time-points, different patients, different tissues (cancer vs. control) etc. Reverse transcription (RT) was organised in the same way, but the set of samples for RT always differed from that one used during isolation. In real-time PCR samples to be compared were measured with the same master mix. To eliminate bath effect in the time-series experiment expression of clock genes per2 and bmal1 was measured (Figure S3A). As clock genes per2 and bmal1 exerts inverted acrophase in their mRNA daily pattern, their expression was used as an internal control.
Minor Comments
— Abstract language.
The abstract is informative but overly dense. Consider simplifying and clearly separating background, methods, results, and conclusions.
- thank you for the comment, we separated background, methods, results, and conclusions and did our best to make abstract more concise and less dense.
— Typographical errors.
There are several minor typographical errors throughout the manuscript (e.g., “camptotecin” instead of “camptothecin”, and inconsistent spacing around punctuation). Please revise carefully.
- thank you for the comment, we checked MS carefully. Unfortunately, inconsistent spacing appears again and again because of different Word versions of people working with the text. We correct it all the time a hope that we found all newly emerged incorrect spacing in the submitted revision.
— Figure legends.
Some figure legends, especially for Figures 7–11, could benefit from more concise and self-contained descriptions of what is being shown and what the main finding is.
- thank you for the comment, all figure legends were corrected accordingly.
— Statistical annotations.
Ensure consistent reporting of statistical results (e.g., p-values, confidence intervals, and n numbers). In some cases, "P" is capitalized, while in others "p" is used.
- thank you for the comment, statistical annotations were controlled and corrected.
— Abbreviations.
Please ensure all abbreviations are defined upon first use (e.g., DLD1, RF-EMF, LD cycle, ZT time).
- thank you for the comment, we carefully controlled all abbreviations including LD, h, s, ZT and others.
In summary, this is a valuable and well-designed study that contributes new insights into the chronobiology of miRNAs and their role in CRC. Addressing the comments above will strengthen the manuscript’s impact and clarity.
Reviewer 2 Report (Previous Reviewer 3)
Comments and Suggestions for Authors
The manuscript presents a broad approach to the analysis of miRNAs. The following points are recommended for improvement:
- The authors mention that the circadian rhythm and age are involved in regulating miRNAs, based on studies using rat colon and DLD1 cells. Similar to what has been observed in human patients, and to further support this concept, does the age of the animals or the passaging of DLD1 cells also have a (comparable) effect?
- More details regarding DLD culture are recommended, accordingly.
- It would be recommendable to include quantitative changes of proteins, or to mention this limitation of the study.
Some editing and corrections appear to be required, e.g. in the introduction, the epithelial barrier mistakenly is named “endothelial barrier”.
Author Response
Dear reviewer,
Thank you very much for your valuable time and opinion, we appreciate it very much. All your suggestions and comments were implemented into the text, please see detailed response bellow.
Changes in the text are indicated by blue background of the text.
Sincerely,
Iveta Herichová on behalf of the all authors
Reviewer 2
Comments and Suggestions for Authors
The manuscript presents a broad approach to the analysis of miRNAs. The following points are recommended for improvement:
- The authors mention that the circadian rhythm and age are involved in regulating miRNAs, based on studies using rat colon and DLD1 cells. Similar to what has been observed in human patients, and to further support this concept, does the age of the animals or the passaging of DLD1 cells also have a (comparable) effect?
- thank you very much for the inspirative questions. Unfortunately, recently it is not possible to provide an exact answer. However, it is known, that aging influences the circadian system functioning and clock gene expression in human and in animals. Experimental evidence usually implicates decrease in functionality of the circadian oscillator with increasing age (Hood and Amir, 2017).
It has been shown that the age of human probands strongly influences clock gene expression in human primary fibroblast culture with cry1 exerting similar trend in expression like in our study (Kalfalah et al., 2016). Cry1 can contribute to cell survival by promoting p53 degradation (Jia et al., 2021). Similarly, expression of miR-150-5p in airways was shown to decrease with increasing age of human probands (Ong et al., 2019).
It is known that passaging strongly influence transcriptome (e.g. Dessel et al., 2019). Aortic vascular smooth muscle cells show a decrease in expression of clock genes per2 and bmal1 after 15-16 passages (Kunieda et al., 2006). Concerning miRNAs, it has been shown that approximately 50 % of miRNAs present in cell culture of adult human lung fibroblasts was sensitive to number of passaging (Ikari et al., 2015).
Passaging of cell culture can be considered as a model of senescence in respect to telomere length and telomerase activity. Telomerase activity has been associated with miRNA expression (Bonifacio et al., 2010).
Therefore yes, it is reasonable to expect that in cancer tissue of senescent animals would miR-150-5p decrease and cry1 increase more than in young animals. If passaging of cell would cause the same effect in respect to miR-150-5p/cry1 expression like in younger and older patients needs to be experimentally tested.
Bonifacio LN, Jarstfer MB. MiRNA profile associated with replicative senescence, extended cell culture, and ectopic telomerase expression in human foreskin fibroblasts. PLoS One. 2010 Sep 1;5(9):e12519. doi: 10.1371/journal.pone.0012519.
Dessels C, Ambele MA, Pepper MS. The effect of medium supplementation and serial passaging on the transcriptome of human adipose-derived stromal cells expanded in vitro. Stem Cell Res Ther. 2019 Aug 14;10(1):253. doi: 10.1186/s13287-019-1370-2.
Hood S, Amir S. The aging clock: circadian rhythms and later life. J Clin Invest. 2017 Feb 1;127(2):437-446. doi: 10.1172/JCI90328.
Jia M, Su B, Mo L, Qiu W, Ying J, Lin P, Yang B, Li D, Wang D, Xu L, Li H, Zhou Z, Li X, Li J. Circadian clock protein CRY1 prevents paclitaxel‑induced senescence of bladder cancer cells by promoting p53 degradation. Oncol Rep. 2021 Mar;45(3):1033-1043. doi: 10.3892/or.2020.7914.
Ikari J, Smith LM, Nelson AJ, Iwasawa S, Gunji Y, Farid M, Wang X, Basma H, Feghali-Bostwick C, Liu X, DeMeo DL, Rennard SI. Effect of culture conditions on microRNA expression in primary adult control and COPD lung fibroblasts in vitro. In Vitro Cell Dev Biol Anim. 2015 Apr;51(4):390-9. doi: 10.1007/s11626-014-9820-8.
Kalfalah F, Janke L, Schiavi A, Tigges J, Ix A, Ventura N, Boege F, Reinke H. Crosstalk of clock gene expression and autophagy in aging. Aging (Albany NY). 2016 Aug 28;8(9):1876-1895. doi: 10.18632/aging.101018.
Kunieda T, Minamino T, Katsuno T, Tateno K, Nishi J, Miyauchi H, Orimo M, Okada S, Komuro I. Cellular senescence impairs circadian expression of clock genes in vitro and in vivo. Circ Res. 2006 Mar 3;98(4):532-9. doi: 10.1161/01.RES.0000204504.25798.a8.
Ong J, Woldhuis RR, Boudewijn IM, van den Berg A, Kluiver J, Kok K, Terpstra MM, Guryev V, de Vries M, Vermeulen CJ, Timens W, van den Berge M, Brandsma CA. Age-related gene and miRNA expression changes in airways of healthy individuals. Sci Rep. 2019 Mar 6;9(1):3765. doi: 10.1038/s41598-019-39873-0.
- More details regarding DLD culture are recommended, accordingly.
Details about DLD1 culture were added into chapter “Materials and Methods” which was modified extensively. Information about DLD1 cell line was added into the chapter “Discussion”:
DLD1 is near-diploid cell line possessing Y chromosome derived from the large intestine of adult male patient diagnosed with poorly or moderately differentiated colon adenocarcinoma. DLD1 cells bear pS241F mutation in the tp53 gene on one allele while the other allele is silent. pS241F mutation causes generation of non-functional p53 protein.
- It would be recommendable to include quantitative changes of proteins, or to mention this limitation of the study.
- thank you for the comment, we agree. We included new statement in the section “Limitations of the study”:
We included following statement in the section “Limitations of the study”:
Recent study does not include evidence that miR-150-5p inhibits protein products of bcl2, myb and cry1. However, inhibitory effect miR-150-5p on bcl2 and myb translation was already demonstrated in CRC cell lines LoVo and HT27 elsewhere [53,55]. High level of correlation between mRNA and protein products biosynthesis of clock gene [197,198,199] and cry1 in particular was also shown [200]. Therefore, we suppose that mRNA based evidence provided in our study is satisfactory for conclusions.
- Oishi, K.; Sakamoto, K.; Okada, T.; Nagase, T.; Ishida, N. Antiphase circadian expression between BMAL1 and period homologue mRNA in the suprachiasmatic nucleus and peripheral tissues of rats. Biochem Biophys Res Commun.1998, 253, 199-203, doi: 10.1006/bbrc.1998.9779.
- Yan, L.; Takekida, S.; Shigeyoshi, Y.; Okamura, H. Per1 and Per2 gene expression in the rat suprachiasmatic nucleus: circadian profile and the compartment-specific response to light. Neuroscience1999, 94, 141-150, doi: 10.1016/s0306-4522(99)00223-7.
- Reppert, S.M.; Weaver, D.R. Molecular analysis of mammalian circadian rhythms. Annu Rev Physiol.2001, 63, 647-676, doi: 10.1146/annurev.physiol.63.1.647.
- van der Watt, P.J.; Roden, L.C.; Davis, K.T.; Parker, M.I.; Leaner, V.D. Circadian Oscillations Persist in Cervical and Esophageal Cancer Cells Displaying Decreased Expression of Tumor-Suppressing Circadian Clock Genes. Mol Cancer Res.2020, 18, 1340-1353, doi: 10.1158/1541-7786.MCR-19-1074.
- just to explain, not included in MS:
Regulation of clock genes occurs mainly at the transcriptional level via E-box in their promoter. Basic feedback loop controls transcription of so called clock controlled genes also via E-box (Reppert and Weaver, 2001). It is known that pronounced changes in clock gene expression are followed by change in corresponding protein level and therefore results based on measurement of clock gene mRNA expression are well accepted in biological rhythm oriented research (e.g. Yan et al., 1999; Oishi et al., 1998).
Reppert SM, Weaver DR. Annu Rev Physiol. 2001; 63:647-76. - 1284 citations in Scopus
Yan L, Takekida S, Shigeyoshi Y, Okamura H. Neuroscience. 1999; 94(1):141-50. - 219 citations in Scopus
Oishi K, Sakamoto K, Okada T, Nagase T, Ishida N. Biochem Biophys Res Commun. 1998; 253(2):199-203. - 204 citations in WOS
Comments on the Quality of English Language
Some editing and corrections appear to be required, e.g. in the introduction, the epithelial barrier mistakenly is named “endothelial barrier”.
- thank you very much for the comment, we agree. Manuscript was carefully controlled for typographical errors, sentence structure and content as was requested.
- thank you very much for the comment, “endothelial barrier” was replace by “epithelial barrier”
Round 2
Reviewer 1 Report (Previous Reviewer 1)
Comments and Suggestions for Authors
I thank the authors for their great effort to improve the article. This reviewer is satisfied with the work. I believe there are no further issues to add on my part.
This manuscript is a resubmission of an earlier submission. The following is a list of the peer review reports and author responses from that submission.
Round 1
Reviewer 1 Report
Comments and Suggestions for Authors
In the article, the authors discuss a possible relationship between circadian regulation of miRNAs and colon cancer progression, using the rat as an experimental model. This is a superficial analysis in which the authors describe the microRNAs found in the colon of rats at different stages of the circadian rhythms, but without an obvious relationship with colon cancer or any other type of cancer.
Major comments
— The paper deals with the role of microRNAs in the context of colon cancer. However, the introduction barely discusses microRNAs in the colon and the role they play in this organ under basal conditions. The authors should add a section in the introduction dealing with this issue.
— Description of the influence or circadian rhythms, microRNAs an cancer in the introduction must be added.
— The authors do not discuss colon cancer in any way in their article. The article is based on a mere descriptor of microRNAs observed in colon, which is neither original nor brings any new message. There are already a multitude of papers in the literature about cycadian rhyrms and colon cancer (some examples are 10.1126/sciadv.abo2389, 10.1038/s41590-024-01859-0, 10.1016/j.gene.2021.145894). Authors must include significant results in the article to gain relevance and be accepted in this journal. They should at least perform the same experiments on rat colon cancer samples and compare the expression of these microRNAs with the control samples. Furthermore, once these microRNAs have been established, they should administer these molecules to rats with colon cancer to observe whether there is a greater or lesser progression of the tumors, or whether they favor metastasis.
In the conditions in which the article is submitted, I find no direct relation with the subject of the journal nor with the proposed title, so I consider that it should not be accepted, unless significant experiments are carried out to increase the originality and quality of the research. The article is based on mere correlations and speculation on the part of the authors, so it cannot be accepted under the conditions in which it is presented.
Reviewer 2 Report
Comments and Suggestions for Authors
Title: Daily profile of miRNAs in the rat colon and their relationship 2 to colorectal cancer
Comments:
1. Is EGF and TGF are the only pathways involved in the cancer type? Since 371 mature and 442 pre-mature miRNA involved in this study, What are the other pathways involved? 2. What are the significance of the reported miRNA in other cancer types? 3. The mentioned miRNAs in table 1 are oncomiRs or Tumor suppressor miRNAs? 4. Which stage of colon cancer is included in this study?
Reviewer 3 Report
Comments and Suggestions for Authors
Herichová et al. have submitted entitled “Daily profile of miRNAs in the rat colon and their relationship to colorectal cancer” for publication in Cancers.
The authors have analyzed the timing of miRNA expression, reporting a peak of miRNA's during the first half of the dark phase, with a trend to higher expression of DGCR8. The authors describe that genes interacting with miRNAs that have peak expression during the dark phase have a greater impact on angiogenesis, the immune system, and EGF and TGF-beta signaling compared to those that peak in the light phase, and additionally, that the target genes of “dark” miRNAs have distinct effects on the Toll and Notch signaling pathways, as well as on the hypoxia response through HIF activation.
The following points a recommended for improvement of the manuscript.
1. The authors describe that the circadian rhythm is involved in the regulation of apoptosis. A direct link to the reported miRNAs might allow a proof of concept.
2. It would be helpful to include quantitative changes of proteins.
3. A separate model or comparison with related tissue might strengthen the study.
4. Trends should be handled with more care, as these may not only be not significant, but also difficult to reproduce.
5. As the study was performed in rats, species-specific miRNAs might not be excluded from the study.
6. Some language editing might be recommendable.
Comments on the Quality of English LanguageSome language editing for better readability recommended.